# ATTENTION-BASED GRAPH CORESET LABELING FOR ACTIVE LEARNING

## ABSTRACT

Graph Convolutional Networks (GCNs) have demonstrated their effectiveness in a variety of graph-based tasks. However, their performance heavily depends on the availability of a sufficient amount of labeled data, which is often costly to acquire in real-world applications. To address this challenge, GNN-based Active Learning (AL) methods have been proposed to improve labeling efficiency by selecting the most informative nodes in a graph for labeling. The existing graph active learning methods employ different heuristic approaches, while efficiency sometimes, they fail to explicitly explore the influence of labeled data on unlabeled data, thus limiting the generalizability of graph models to various types of graph data. In this paper, we propose an Attention-based Graph Coreset Labeling framework (AGCL). AGCL can, with limited budgets, gradually discover core data to be labeled from a global view so as to obtain a training dataset that can efficiently depict the whole graph space and maximize the performance of GNNs. Specifically, we explicitly explore and exploit the correlations between nodes in the unlabeled pool and those in the labeled pool using an attention architecture and directly connect the correlations with the prediction performance on unlabeled set. By leveraging influence (attention) scores, AGCL identifies and labels data with the maximum representation difference from the existing labeled pool, thereby enhancing sample complexity. We theoretically demonstrate the superiority of the attention-based data selection strategy in reducing the covering radius bound, thereby improving the expected prediction performance on unlabeled data. Our experimental results show that the labeled coreset significantly enhances the generalizability of various graph models across different graph datasets, as well as CNN models in image classification tasks.

## 1 INTRODUCTION

Graph neural networks (GNNs) (Duvenaud et al. (2015); Kipf & Welling (2017); Hamilton et al. (2017)) have emerged as powerful approaches for learning representations of graph-structured data. It has been noted (Zhang et al. (2022b)) that success of GNNs in various graph-based learning tasks (Xu et al. (2018); Klicpera et al. (2019); Zhang & Chen (2018)) requires plenty of labeled data. However, sufficient informative training data is often not available, as human annotation is expensive and time-consuming, particularly for biological graphs that contain specialized structures requiring expert labeling.

Active learning (AL) provides solutions by selecting and annotating a few highly informative and representative points that can depict a large portion of the data space, especially uncertain regions. While various active learning methods have been proposed and applied for CNN models (Sener & Savarese (2018); Caramalau et al. (2021); Wang et al. (2016); Yoo & Kweon (2019)) on independent and identically distributed (i.i.d) data, these methods fail to capture both the graph structure and node features, leading to suboptimal performance when applied to GNNs (Gao et al. (2018a); Madhawa & Murata (2020)). Additionally, the interconnected and interdependent nature of nodes in a graph means that the choice of labeled data partitions has a significant impact on the performance of GNN models (Shchur et al. (2018); Fu et al. (2024)). Therefore, it is not directly applicable to apply active learning methods from CNNs to GNNs.

To select more representative data on graphs, GNN-based active learning methods (Gao et al. (2018b); Cai et al. (2017); Wu et al. (2019)) incorporate graph structural information into query heuristics (uncertainty, diversity, or density). Several recent works (Zhang et al. (2022b; 2021e;a;c)) considered the characteristic of influence propagation in the graph and proposed a series of graph active learning methods aiming to identify nodes with maximum influence for the rest. For example, the Grain method (Zhang et al. (2021e)) connects labeled data selection in GNNs with social influence, maximizing the number of unlabeled nodes influenced by labeled ones. The intuition behind these methods is to exploit the assumption that nodes that are close in feature space and graph structure are likely to have the same label, i.e., they focus on the local structure of the graph.

While efficient, these strategies 1) lack a direct correlation with the expected prediction performance on unlabeled nodes in the final task, and 2) mainly focus on the local graph structure, failing to comprehensively explore the influence between labeled and unlabeled data across the entire graph space. However, in real-world applications, graphs can be complex; for instance, in heterophilic graphs, connected nodes may have different labels. These challenges raise a critical question for graph annotation: *Given a fixed labeling budget, how can we develop a general framework that efficiently and effectively identifies core data in the graph by considering both the graph structure (local and global) and features, ultimately improving model performance?*

In this paper, we propose a general graph active learning framework called Attention-based Graph Coreset Labeling (AGCL). We address the graph annotation problem as an unlabeled coreset selection problem for GNNs, focusing on selecting data that maximizes coverage of the remaining data in the graph representation space. The challenge of graph coreset selection lies in designing an effective measure that evaluates the correlations between labeled and unlabeled data, while considering the complex graph structure and features, which directly links to the expected predictions on unlabeled data. In AGCL, we explicitly connect the labeled and unlabeled pools beyond the original graph connections, construct the influence between them using an attention-aggregation strategy in the embedding space, iteratively select core data from a global perspective. We theoretically demonstrate that selecting unlabeled data with the maximum representation difference from the labeled pool, based on an attention-based metric, helps reduce the bound radius $\delta$, thereby decreasing the total loss of the graph data. Empirically, we demonstrate the effectiveness of the proposed method across various GNN architectures and different types of graph data (homophily and heterophily) as well as different data scales (same-scale and large-scale). Additionally, we illustrate how AGCL can serve as a general active learning framework, extending its applicability to image classification. In summary, our main contributions are:

- We propose an attention-based active learning framework for graph models, which iteratively selects and annotate data in a graph by addressing the coreset selection problem for non-i.i.d. graph data.

- We theoretically prove the superiority of the attention-based selection strategy: selecting unlabeled data with the maximum representation difference from the current labeled pool can help reduce the bound in graph coreset selection and directly enhance the performance of the graph model.

- Our proposed AGCL is a general active learning framework that can be applied to both graph data and image tasks. We conduct extensive experiments on both types of data to demonstrate the effectiveness of the proposed method for various classification tasks.

## 2 RELATED WORKS

### 2.1 GRAPH NEURAL NETWORKS

In recent years, graph neural networks (GNNs) have attracted increasing attention due to their superiority in the processing of graph-structured data (Henaff et al. (2015); Kipf & Welling (2017); Gilmer et al. (2017); Bronstein et al. (2017); Velickovic et al. (2018)). To improve the expressive power of GNNs, different message-passing schemes have been developed to propagate and aggregate neighborhood information (Kipf & Welling (2017); Velickovic et al. (2018); Feng et al. (2020)). Recently, some studies tried to understand the generalization ability of GNNs from the perspective of training data. Zhu et al. (2021) explored the influence of training data and presented Shift-Robust GNN (SR-GNN), designed to account for distributional differences between biased training data and

a graph's true inference distribution. Ma et al. (2021) extended PAC-Bayesian analysis for graph data to analyze the generalization performance of GNNs, and demonstrated that the distance between a test subgroup and the training set can be a key factor affecting the GNN performance. Su et al. proved that the distance of the training set to the rest of the vertexes in the graph is negatively correlated to the learning outcome of GNNs.

## 2.2 ACTIVE LEARNING ON GRAPHS

In practice, obtaining sufficient informative training data is challenging, as human annotation is expensive and time-consuming. Active learning and semi-supervised representation learning with few labels are both designed to address the scarcity of labeled data, but from different perspectives. While few-labeled semi-supervised learning focuses on comprehensively leveraging the small amount of labeled data and the large amount of unlabeled data to achieve better performance, active learning focuses on selecting and labeling the most informative nodes to maximize model performance with minimal cost.

Generally, active learning is an iterative labeling process in which a model is learned at every iteration, and a set of data points is chosen to be labeled from a pool of unlabelled points to maximize model performance. Based on the query strategy, the majority of work can be divided into three categories (Settles (2009)): theoretically-motivated methods (MacKay (1992)), ensemble approaches (McCallum et al. (1998); Freund et al. (1997)) and uncertainty based (Tong & Koller (2001); Li & Guo (2013); Settles & Craven (2008)). Demir et al. (2010) used a heuristic to first filter the pool based on uncertainty and then choose the points to label using diversity. Sener & Savarese (2018) proposed an effective batch active learning method for deep CNNs. In this method, the active learning problem is defined as coreset selection; however, it is only for nonstructural data. Several attempts have been made for applying AL on graph-structured data (Bilgic et al. (2010); Gu et al. (2013); Kuwadekar & Neville (2011)) based on a graph signal processing framework. Subsequently, a series of GNN-based AL methods (Cai et al. (2017); Gao et al. (2018a)) have been studied using different metrics, including uncertainty, information density, and graph centrality to evaluate training data. However, simply combining these metrics may not select informative data. Recently, several works (Cui et al. (2022); Ma et al. (2022); Zhang et al. (2022a); Fuchsgruber et al. (2024); Wu et al. (2019); Li et al. (2020); Zhang et al. (2021c)) were proposed to further consider the graph information. To maximize the coverage of the labeled data, a new node selection metric is proposed in ALG (Zhang et al. (2021a)) to maximize the effective reception field. Grain (Zhang et al. (2021e)) further generalizes the reception field to the number of activated nodes in social influence maximization. Reinforcement learning (Hu et al. (2020a)) and LLM (Chen et al. (2023)) are also used to improve active learning on graphs. While most existing approaches are based on some query heuristics to implicitly encode the relationships between labeled and unlabeled data, they often struggle to identify truly informative data points in the face of complicated graph structures.

## 2.3 CORESET SELECTION

Coresets are defined as small and informative weighted data subsets, ensuring that models fitted to the coreset also provide a good fit for the original data. Several works, such as those by Wei et al. (2015), Mirzasoleiman et al. (2020), and Killamsetty et al. (2021a), have studied the efficient training of deep learning models using selected coresets. Mirzasoleiman et al. (2020) focused on selecting representative coresets of the training data that closely estimate the full training gradient. Killamsetty et al. (2021b) treated coreset selection as an optimization problem for the validation set loss, aiming for efficient learning with a focus on generalization. Killamsetty et al. (2021a) proposed GRAD-MATCH, which selects subsets approximating the full training loss or validation loss gradient using orthogonal matching pursuit. Meanwhile, coreset selection methods (Sener & Savarese (2018); Ash et al. (2019)) were also used for active learning scenarios, where a subset of data instances from the unlabeled set is selected to be labeled.

# 3 PRELIMINARIES

In this section, we formally define the problem of active learning for GNNs under the semi-supervised node classification setting.

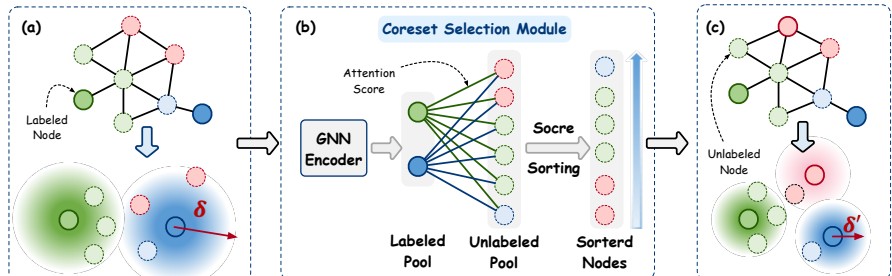

Figure 1: (a) The input graph and a visualization of the influence of labeled data with bound $\delta$ in embedding space. (b) AGCL explicitly construct the influence of data in unlabeled pool for labeled data by an attention-based networks. The unlabeled data which have minimum representation influence are selected into labeled pool. (c) The output graph with selected informative data and a visualization of the decreased bound $\delta$.

We are given a graph $\mathcal{G} = (\mathcal{V}, \mathcal{E})$ with the node set $\mathcal{V}$ and edge set $\mathcal{E}$. Suppose there are $N$ nodes in $\mathcal{V}$ and each node $v_i \in \mathcal{V}$ has an associated feature vector $\boldsymbol{x}_i \in \mathbf{X} \in \mathbb{R}^{N \times d}$ and a label vector $y_i \in \mathbf{Y} \in \{0,1\}^{N \times C}$. The connection among nodes can be described by the adjacency matrix $\mathbf{A}$, with $\mathbf{A}_{ij} = 1$ if there exists an edge $(v_i, v_j)$, otherwise $\mathbf{A}_{ij} = 0$. Here, we focus on the $C$ class node classification task on graph $\mathcal{G}$, with a label space $\mathcal{Y} = \{1, \dots, C\}$.

In active learning on graphs, we consider that the full node set $\mathcal{V}$ is partitioned into training set $\mathcal{V}_{train}$, validation set $\mathcal{V}_{val}$, and test set $\mathcal{V}_{test}$. The training set $\mathcal{V}_{train}$ contains labeled and unlabeled data. An active learning algorithm $A_{\mathbf{s}}$ iteratively selects extra data from the unlabeled pool $\mathcal{V}_u$ and gives labels to them into a labeled pool $\mathcal{V}_l$. With a labeling budget $\mathcal{B}$ and an initial labeled pool $\mathbf{s}^0 = \left\{ s_j^0 \in \mathcal{V}_{train} \right\}_{j<m}$, an active learning algorithm expects to minimize the future expected loss with a GNN model $\mathcal{M}$ by:

$$\min_{\mathbf{s}^{k+1}:|\mathbf{s}^{k+1}|\leq\mathcal{B}} E_{\boldsymbol{x},\boldsymbol{y}} \left[ l_{\mathcal{M}} \left( \mathcal{G}, \boldsymbol{x}, y; A_{\mathbf{s}^0 \cup \dots \mathbf{s}^{k+1}} \right) \right]. \tag{1}$$

## 4 ATTENTION-BASED GRAPH CORESET LABELING (AGCL)

In this section, we present AGCL, a general graph active learning framework using a labeling coreset from the graph data to maximize the generalization ability of various GNN models. We define graph annotation as a coreset selection problem without labels on GNNs in subsection 4.1. To address this problem, we theoretically showed that labeling the unlabeled data that have maximum representation difference compared to the existing labeled pool obtains a smaller bound radius $\delta$ and reduce the total prediction loss in subsection 4.2. Hence we construct an attention-based model to explicitly evaluate the influence of each unlabeled sample $u$ for each labeled sample $v$. While capturing the correlations between labeled and unlabeled data from a global perspective, AGCL selects and labels data according to attention scores at each batch of data labeling, enhancing the information complexity in the labeled pool. The above process is repeated until the labeling budget $\mathcal{B}$ runs out. We introduce each component of AGCL in subsection 4.3.

### 4.1 CORESET SELECTION ON GRAPHS

Generally, the active learning algorithm acquires one label at a time by querying the oracle in each iteration, *i.e.*, $\mathcal{B} = 1$. However, for graphs containing a large number of nodes and edges, this is infeasible. Therefore, we focus on the batch active learning technique (Contardo et al. (2017)) in which the active learning algorithm chooses a set of data to be labeled by an oracle at each iteration. In a classification problem, given a training set, the loss of the training set can be calculated as:

$$R_{emp} = \frac{1}{|\mathbf{s}|} \sum_{i \in \mathbf{s}} l_{\mathcal{M}} \left( \mathcal{G}, x_i, y_i; A_{\mathbf{s}} \right), \tag{2}$$

where $|\mathbf{s}|$ is the number of labeled points and the empirical risk $R_{emp}$ is the average loss of all training samples. After training a GNN model $\mathcal{M}$, the aim is to predict the outputs for new or unseen data. Among the generated hypotheses, the best hypothesis is the one that minimizes the expected value of the loss over the whole input space, which is defined as:

$$R = \frac{1}{N} \sum_{j \in [N]} l_{\mathcal{M}} \left( \mathcal{G}, x_j, y_j; A_{\mathbf{s}} \right). \tag{3}$$

When designing an active learning algorithm method for GNNs, the goal is to minimize the generalization gap between $R_{emp}$ and $R$:

$$\min_{\mathbf{s}^1 : |\mathbf{s}^1| \leq \mathcal{B}} \left| \frac{1}{N} \sum_{i \in [N]} l_{\mathcal{M}} \left( \mathcal{G}, x_i, y_i; A_{\mathbf{s}^0 \cup \mathbf{s}^1} \right) - \frac{1}{|\mathbf{s}^0 + \mathbf{s}^1|} \sum_{j \in \mathbf{s}^0 \cup \mathbf{s}^1} l_{\mathcal{M}} \left( \mathcal{G}, x_j, y_j; A_{\mathbf{s}^0 \cup \mathbf{s}^1} \right) \right|, \tag{4}$$

where $[N] = \{1, \ldots, N\}$. In other words, given the initial labeled set $\mathbf{s}^0$ and the budget $\mathcal{B}$, we aim to find a set of points to query labels $\mathbf{s}^1$ such that when we train a GNN model, the performance of the model on the labeled subset is as close as possible to its performance on the entire dataset.

## 4.2 THEORETICAL ANALYSIS

The optimization objective equation 4 is not directly computable since we do not have access to all the labels. In (Sener & Savarese (2018)), an upper bound is given to the objective function of coreset on CNNs. As shown in Theorem 1 Sener & Savarese (2018), we can bound this loss with covering radius $\delta$., i.e., $\left| \frac{1}{N} \sum_{i \in [N]} l \left( \mathbf{x}_i, y_i, A_{\mathbf{s}} \right) - \frac{1}{|\mathbf{s}|} \sum_{j \in \mathbf{s}} l \left( \mathbf{x}_j, y_j; A_{\mathbf{s}} \right) \right| \leq \mathcal{O}\left( \delta_{\mathbf{s}} \right) + \mathcal{O}\left( \sqrt{\frac{1}{N}} \right)$ where $\delta_{\mathbf{s}}$ with radius $\delta$ centered at each labeled sample in $s$ can cover the entire representation space. Obviously, if we want to reduce the loss, we need to decrease the covering radius.

Although this bound provides the original analysis of coreset selection, it is also important for directly analyzing the influence of training/labeled data on the prediction performance of testing/unlabeled data, offering theoretical guarantees for AL performance, particularly in the context of graph AL.

**Proposition 4.1** *Given a graph $\mathcal{G}$, for any labeled data $v$ with the hidden representation $\boldsymbol{h}_v$, there exist a $\delta_v > 0$, such that for two unlabeled nodes $\{u, u'\}$ with representations $\boldsymbol{h}_u$ and $\boldsymbol{h}_{u'}$, if the distance $\Delta(u, v) < \Delta(u', v) < \delta_v$, then $l(f(\boldsymbol{h}_u)) < l(f(\boldsymbol{h}_{u'}))$, where $f(\cdot)$ is the prediction function, and $l(\cdot)$ is the loss function.*

This proposition states that for each training sample $v$, its hidden representation $\boldsymbol{h}_v$ has a $\delta_v$ cover in the embedding space. The prediction performance (loss) of two samples whose embedding located in the radius $\delta_v$ centered at $v$ admits simple monotonicity with respect to their distance to sample $v$. The visualization about covering radius $\delta$ can be found in Figure 1 (a). Proposition 4.1 is formulated for abstract points in the embedding space, making it applicable to both i.i.d. data and non-i.i.d. graph data. For graph data, the embedding space encodes the complex graph connections.

Extending this to the entire graph data, we can conclude that a GNN trained on a training set with closer distances to the remaining data—indicating an approximate coverage of the whole representation space with a smaller covering radius—exhibits better performance in Lemma 4.1.

**Lemma 4.1** *Assume there are two training set $\mathcal{V}_{train}$ and $\mathcal{V}'_{train}$, and test set $\mathcal{V}_{test}$. Based on two training set, we get two trained GNN models $\mathcal{M}(\mathcal{V}_{train})$ and $\mathcal{M}(\mathcal{V}'_{train})$. If $\sum_{v \in \mathcal{V}_{train}} d(v, \mathcal{V}_{test}) < \sum_{u \in \mathcal{V}'_{train}} d(u, \mathcal{V}_{test})$, thus the covering radius $\delta_{v \in \mathcal{V}_{train}} < \delta_{u \in \mathcal{V}'_{train}}$, we have $\sum_{u \in \mathcal{V}_{test}} l(f_{\mathcal{M}(\mathcal{V}_{train})}(u)) < \sum_{u \in \mathcal{V}_{test}} l(f_{\mathcal{M}(\mathcal{V}'_{train})}(u))$.*

To tackle coreset selection on graph, it is important to explicitly show the relationship (distance) of unlabeled data and labeled data through a measure that relates to final prediction performance.

**Lemma 4.2** *Assume a graph has a labeled data, $v$, with the hidden representation $\boldsymbol{h}_v$, and two unlabeled nodes, $\{u_1, u_2\}$, with representations $\{\boldsymbol{h}_{u_1}, \boldsymbol{h}_{u_2}\}$. If $\boldsymbol{h}_v = \alpha_1 \boldsymbol{h}_{u_1} + \alpha_2 \boldsymbol{h}_{u_2}$, where the score $\alpha_1 > \alpha_2 \approx 0$, then selecting $u_2$ into the labeled pool (as $u_2 \to v_2$) results in a smaller total loss across the entire graph space than selecting $u_1$ into the labeled pool (as $u_1 \to v_1$).*

*Proof.* Consider a labeled input $v$, and two unlabeled inputs $u_1$ and $u_2$. If the hidden representation of $v$ can be represented as $\boldsymbol{h}_v = \alpha_1 \boldsymbol{h}_{u_1} + \alpha_2 \boldsymbol{h}_{u_2}$, where $\alpha_1 > \alpha_2 \approx 0$, i.e., $d(v, u_1) < d(v, u_2) \leq \delta_v$ where $\delta_v$ is the covering radius of point $v$. The loss of $v$ is then given by $l(\boldsymbol{h}_v)$, where $l(\cdot)$ is the loss function. For simplicity, we assume zero training loss (see that in Assumption 2 in Appendix A.2), leading to: $l(\boldsymbol{h}_v) = 0$.

Selecting the data $u_2$ that has the maximum difference with $v$ according to scores $\{\alpha_1, \alpha_2\}$ and adding it to the labeled pool as $u_2 \to v_2$, we now have two training nodes, $v$ and $v_2$, and one testing node, $u_1$. The total loss $L_1$ on the entire input space is: $L_1 = l(\boldsymbol{h}_v) + l(\boldsymbol{h}_{v_2}) + l(\boldsymbol{h}_{u_1}) \approx l(\boldsymbol{h}_{u_1})$ since the training loss on the training set is zero. From the perspective of the covering radius, the loss $l(\boldsymbol{h}_{u_1}) \leq \mathcal{O}(\delta_1) \leq \mathcal{O}(\max(d(v, u_1), d(v_2, u_1)))$.

Consider the scenario in which $u_1$ is selected to the labeled pool, we get the loss $L_2 = l(\boldsymbol{h}_v) + l(\boldsymbol{h}_{v_1}) + l(\boldsymbol{h}_{u_2}) \approx l(\boldsymbol{h}_{u_2})$, $l(\boldsymbol{h}_{u_2}) \leq \mathcal{O}(\delta_2) \leq \mathcal{O}(\max(d(v, u_2), d(v_1, u_2)))$.

As $d(v, u_1) < d(v, u_2)$ and $d(v_2, u_1) = d(v_1, u_2)$, we have $\max(d(v, u_1), d(v_2, u_1)) < \max(d(v, u_2), d(v_1, u_2))$, thus, $\delta_1 < \delta_2$. According to Proposition 4.1, we have $l(\boldsymbol{h}_{u_1}) < l(\boldsymbol{h}_{u_2})$, $L_1 < L_2$. $\square$

Lemma 4.2 clearly demonstrates the benefit of selecting the coreset based on an explicit metric (representation scores) between the labeled and unlabeled pools in improving expected prediction accuracy.

With Lemma 4.1 and 4.2, we establish a connection between the core data selection and prediction loss on GNNs, leading to the following theorem.

**Theorem 4.1** *Given the whole sample $\mathcal{V}$ drawn from $\mathcal{G}$, let $\mathcal{V}_l$ represents the labeled pool consisting of points with labels, and let $\mathcal{V}_u$ denotes the set of unlabeled data. $\exists s \in \mathcal{V}_u : \forall v \in \mathcal{V}_l, A_{s,v} < A_{k,v}$ with $k \in \mathcal{V}_u \setminus \{s\}$, where $A_{i,j}$ measures the representation similarity between nodes $i$ and $j$, the larger $A_{i,j}$, the closer nodes $i$ and $j$ are. Thus, we have $\delta_{v \in \mathcal{V}_l \cup \{s\}} < \delta_{v \in \mathcal{V}_l \cup \{k\}} < \delta_{u \in \mathcal{V}_l}$, such that $\sum_{i \in \mathcal{V}} l(f_{\mathcal{M}(\mathcal{V}_l \cup \{s\})}(i)) < \sum_{i \in \mathcal{V}} l(f_{\mathcal{M}(\mathcal{V}_l \cup \{k\})}(i)) < \sum_{i \in \mathcal{V}} l(f_{\mathcal{M}(\mathcal{V}_l)}(i)).$*

Theorem 4.1 indicates that node $s$ is the most informative data point in $\mathcal{V}_u$ with respect to the the existing labeled pool $\mathcal{V}_l$.

In this paper, an attention-based message-passing strategy is proposed to obtain $A_{i,j}$ for coreset selection on graph. In addition to considering the local structure in the graph, attention-based networks also take into account global structural information. By learning the influence between one node in the labeled pool and another in the unlabeled pool and mapping it to the attention matrix, we can intuitively select nodes in the unlabeled pool that are farthest from the current labeled pool to reduce the prediction loss.

### 4.3 ATTENTION-BASED MESSAGE-PASSING AND DATA SELECTION

As illustrated in Figure 1 (b), we design an attention-based graph coreset labeling method to effectively identify the unlabeled data with minimum representation influence to improve the generalization ability of the model.

To obtain the correlations between the labeled and the unlabeled pool, we first learn the hidden representation of nodes by GNN layers to encode the structural information in a graph. Let $\mathbf{X} = [\boldsymbol{x}_1, \boldsymbol{x}_2, \ldots, \boldsymbol{x}_N]^T \in \mathbb{R}^{N \times d}$ be the node features, the $l^{th}$ layer GNN is given by:

$$\boldsymbol{a}_v^{(l)} = \text{Aggregation}^{(l)} \left( \left\{ \boldsymbol{h}_u^{(l-1)} : u \in \mathcal{N}(v) \right\} \right), \boldsymbol{h}_v^{(l)} = \text{Update}^{(l)} \left( \boldsymbol{h}_v^{(l-1)}, \boldsymbol{a}_v^{(l)} \right), \quad (5)$$

where $\boldsymbol{h}_v^{(l)}$ is the hidden feature vector of node $v$ at the $l^{th}$ layer. We initialize $\boldsymbol{h}_v^0 = \boldsymbol{x}_v$, and $\mathcal{N}(v)$ is a set of nodes connected to $v$. We call Aggregation$(\cdot)$ an aggregation function and Update$(\cdot)$ an update function. For example, the layer-wise message-passing in GCN (Kipf & Welling (2017)) is defined as $\boldsymbol{h}_v^{(l)} = \text{ReLU} \left( \mathbf{W} \cdot \text{MEAN} \left\{ \boldsymbol{h}_u^{(l-1)}, \forall u \in \mathcal{N}(v) \cup \{v\} \right\} \right)$, where $\mathbf{W}$ is a layer-specific trainable weight matrix. We get the hidden representation of all labeled nodes with $\boldsymbol{h}^{(l)} = [\boldsymbol{h}_1^{(l)}, \ldots, \boldsymbol{h}_N^{(l)}] \in \mathbb{R}^{N \times d'}$ in the layer of $l$.

Then, we aim to assess the influence of unlabeled data for the existing labeled dataset from a global perspective. Specifically, we employ an attention architecture to explicitly model the relationships between the labeled and unlabeled data pools in the representation space. Assume the labeled pool $\mathcal{V}_l = \{v_1, \ldots, v_m\}$ with features $\boldsymbol{h}^{v\,(l)} = [\boldsymbol{h}_1^{v\,(l)}, \ldots, \boldsymbol{h}_m^{v\,(l)}]$ and unlabeled pool $\mathcal{V}_u = \{u_1, \ldots, u_n\}$ with features $\boldsymbol{h}^{u\,(l)} = [\boldsymbol{h}_1^{u\,(l)}, \ldots, \boldsymbol{h}_n^{u\,(l)}]$ in $l^{th}$ layer. Consider the global information in the graph, for each labeled data $v_i$, we expect to represent it by aggregating the information from the unlabeled pool. Connecting $v_i$ with all unlabeled data in $\mathcal{V}_u$, the hidden representation $\boldsymbol{h}_i^v$ can be obtained by the labeled node $v_i$ acting as the query $\boldsymbol{q}_i^v$ with $\boldsymbol{q}_i^{v\,(l)} = \boldsymbol{h}_i^{v\,(l-1)}\mathbf{W}^Q$:

$$
\begin{aligned}
A_i^{s\,(l)} &= \alpha \boldsymbol{q}_i^{v\,(l)} \mathbf{K}_{\mathcal{V}_u}^{\top}, \\
\boldsymbol{h}_i^{v\,(l)} &= \mathrm{softmax}\left(A_i^{s\,(l)}\right) \mathbf{V}_{\mathcal{V}_u},
\end{aligned}
\tag{6}
$$

where $\alpha$ is a constant scalar ($\alpha = \frac{1}{\sqrt{d'}}$), $\mathbf{K}_{\mathcal{V}_u} = \boldsymbol{h}^u \mathbf{W}^K$ and $\mathbf{V}_{\mathcal{V}_u} = \boldsymbol{h}^u \mathbf{W}^V$ are the key and value matrices of unlabeled pool, respectively. In this way, each labeled node aggregates the information from all unlabeled data in $\mathcal{V}_u$, and the attention score $A_i^{s\,(l)}$ measures the importance of samples in the unlabeled pool for labeled data $v_i$ in representation space.

Similarly, viewing each unlabeled node $u_i$ as query $\boldsymbol{q}_i^u$, its hidden representation can be achieved by aggregating the information from all labeled data:

$$
\boldsymbol{q}_i^{u\,(l)} = \boldsymbol{h}_i^{u\,(l-1)}\mathbf{W}^Q, \boldsymbol{h}_i^{u\,(l)} = \mathrm{softmax}\left(\alpha \boldsymbol{q}_i^{u\,(l)} \mathbf{K}_{\mathcal{V}_v}^{\top}\right) \mathbf{V}_{\mathcal{V}_v},
\tag{7}
$$

where $\mathbf{K}_{\mathcal{V}_v} = \boldsymbol{h}^v \mathbf{W}^K$ and $\mathbf{V}_{\mathcal{V}_v} = \boldsymbol{h}^v \mathbf{W}^V$ are the key and value matrices of labeled pool, respectively.

The equation 6 and equation 7 indicate the computation on single-head attention. In practice, AGCL adopts multi-head attention (MHA) followed by feed-forward blocks (FFN) and layer normalization (LN($\cdot$)) as:

$$
\boldsymbol{h}'^{(l)} = \mathrm{LN}\left(\mathrm{MHA}\left(\boldsymbol{h}^{(l-1)}\right)\right) + \boldsymbol{h}^{(l-1)}; \boldsymbol{h}^{(l)} = \mathrm{LN}\left(\mathrm{FNN}\left(\boldsymbol{h}'^{(l)}\right)\right) + \boldsymbol{h}'^{(l)},
\tag{8}
$$

where $\boldsymbol{h}^{(l)}$ is the representation of labeled and unlabeled data in $l^{th}$ layer. In addition, we incorporate positional encoding, including random walk positional encoding (Dwivedi & Bresson (2021)) and Laplacian positional encoding (Dwivedi et al. (2021)), which are crucial components in transformers, into our proposed AGCL.

**Data selection.** According to Theorem 4.1, to reduce the total loss total input data, we need to select nodes in the unlabeled pool that have the maximum representation difference to the nearest labeled data. Intuitively, data with the smallest similarity to the existing labeled data in the representation space will help maximize sample diversity and complexity. Based on the AGCL algorithm, the attention matrix $A^s$ has explicitly show the importance of nodes in the unlabeled pool for labeled data, thus, we sample node by:

$$
u = \arg\max_{u \in \mathcal{V}_u} \min_{v \in \mathcal{V}_v} A_{v,u}^s.
\tag{9}
$$

Then, we get the labeled pool $\mathcal{V}_l = \mathcal{V}_l \cup \{u\}$. The whole computation process and the complexity analysis of AGCL can be found in Appendix A.1.

## 5 EXPERIMENTS

We conduct experiments to verify that the labeled data selected by our proposed AGCL can enhance the generalization of different graph models. We focus on five popular GNN models: GCN (Kipf & Welling (2017)), GAT (Velickovic et al. (2018), APPNP Klicpera et al. (2019)), H2GCN (Zhu et al. (2020)) and GPRGNN (Chien et al. (2021)). The framework is adaptable to any general GNN model; additional results using GraphSAGE (Hamilton et al. (2017)) are provided in Appendix A.5. Our results show that the core graph data identified by our method can achieve improved performance regardless of the GNN architecture. Additionally, we apply AGCL to image classification tasks

to demonstrate the generalizability of the proposed method. An ablation study on how positional encoding in the self-attention framework affects the performance of AGCL can be found in Appendix A.8. Further robustness analysis of AGCL with noisy data is provided in Appendix A.9.

**Datasets.** Focusing on semi-supervised node classification, we experiment on a range of graph benchmarks: (1) homophilic graph datasets (Cora, Citeseer, Pubmed, and ogbn-arxiv) (Pei et al. (2020); Hu et al. (2020b)) and (2) heterophilic graph datasets (Actor, Squirrel, roman-empire, Penn94) (Zhu et al. (2020); Platonov et al. (2023); Lim et al. (2021)) involving diverse domains and sizes (roman-empire, Penn94 and ogbn-arxiv are large-scale datasets). We also perform experiments on CIFAR-10 (Krizhevsky et al. (2009)) and FashionMNIST (Xiao et al. (2017) Griffin et al. (2007)) datasets for image classification in Appendix A.7. The details of these datasets are provided in Appendix A.4.

## 5.1 EXPERIMENTAL SETTING

We compare our proposed method with other active learning approaches for graphs: Random, Standard, FeatProp (Wu et al. (2019)), AGE (Cai et al. (2017)), GRAIN (Zhang et al. (2021e)), RIM (Zhang et al. (2021d)), ALG (Zhang et al. (2021b)), GraphPart (Ma et al. (2022)), and NC-ALG (Zhang et al. (2024)). For active learning on images, we compare our method with baselines including Random sampling, CoreSet (Sener & Savarese (2018)), VAAL (Sinha et al. (2019)), and (CoreGCN Caramalau et al. (2021)). In general active learning, the initial pool is usually uniformly randomly selected from the whole data. For i.i.d. data, this initial data selection method is reasonable. However, for non-i.i.d. graphs in which nodes are connected by edges, it is of great importance to utilize initial knowledge of the graph. Thus, except for the random selection, we propose a structure and feature-based initial pool selection method.

Considering both the features and graph structure, we propagate features among nodes with the layer-wise propagation rule:

$$\boldsymbol{H}^{(l+1)} = \hat{\mathbf{A}}\boldsymbol{H}^{(l)}, \tag{10}$$

where $\hat{\mathbf{A}} = \hat{\mathbf{D}}^{-1/2}(\mathbf{A}+\mathbf{I})\hat{\mathbf{D}}^{-1/2}$ is a symmetric normalized adjacency matrix, $\mathbf{I}$ is the identity matrix, $\hat{\mathbf{D}}$ is the corresponding degree matrix of $\mathbf{A}+\mathbf{I}$, and $\boldsymbol{H}^{(l)}$ is the hidden node representation in $l^{th}$ layer with $\boldsymbol{H}^{(0)} = \mathbf{X}$. After $k$ iterations of aggregation, the representation of a node $h_i^k$ captures the structural information within its $k$-hop neighborhood. Then, we select $k$ nodes into the initial pool using the k-medoids method.

Our method introduces several hyperparameters, including the number of initial labels $|s^0|$, batch budget $b$, and final labeling budget $\mathcal{B}$. For a fair comparison, we set the final amount of core data obtained for all active learning methods to equal the standard training set for the Cora, Citeseer, and Pubmed datasets. For the ogbn-arxiv dataset, the labeling budget is set to 800, and for the heterophilic datasets, the labeling budget is 600. For $|s^0|$ and $b$, we perform a hyperparameter search for each dataset. For other hyperparameters used in our experiments, including the learning rate, early stopping patience, hidden layer size, dropout rates of the input layer and hidden layer, we usually adopt a similar setting as in Kipf & Welling (2017); Velickovic et al. (2018); Klicpera et al. (2019). Furthermore, all the experiments are conducted on a Linux server equipped with NVIDIA A100. The detailed parameters used in the experiments are listed in Appendix A.3.

## 5.2 RESULTS ON GRAPH TASKS

We conducted experiments on active learning for semi-supervised node classification on homophilic datasets. From Table 1, we can observe that our proposed AGCL outperforms other methods across different graph datasets. Specifically, GCN with the training set selected by AGCL demonstrates improvements of approximately 1.9% and 2.1% over the model trained on the training set selected by GRAIN on Cora and Citeseer, respectively. The effectiveness of AGCL extends to other GNN models, including GAT (Velickovic et al. (2018)) and APPNP (Klicpera et al. (2019)). We further evaluate the influence of the labeling budget, and report the test accuracy of the GCN model versus the number of labeled nodes for training in Appendix 2. Compared with the other baselines, AGCL quickly boosts its accuracy at the beginning of the training and consistently outperforms the baselines as the number of labeled nodes increases. Specifically, to achieve an accuracy of approximately 70% on Citeseer, AGCL requires labeling only 40 samples, whereas other methods need over 60 nodes.

Table 1: Classification accuracy (%) on three citation datasets with different training sets (mean accuracy (%) and standard deviation over 5 different runs).

| Methods | Training Data | Cora | Citeseer | pubmed |
|---|---|---|---|---|
| GCN | Random | 79.81 ± 1.73 | 70.24 ± 2.04 | 76.54 ± 2.60 |
| | Standard | 82.31 ± 0.47 | 71.45 ± 0.69 | 79.59 ± 0.41 |
| | AGE | 80.95 ± 1.14 | 70.34 ± 7.01 | 79.50 ± 2.69 |
| | FeatProp | 77.3 ± 1.36 | 64.0 ± 3.21 | 73.2 ± 1.94 |
| | GRAIN | 80.96 ± 0.40 | 70.96 ± 0.42 | 79.94 ± 0.33 |
| | RIM | 81.78 ± 0.52 | 72.45 ± 0.70 | 76.04 ± 0.83 |
| | ALG | 83.01 ± 0.28 | 71.8 ± 0.09 | 78.52 ± 0.04 |
| | GraphPart | 82.50 ± 0.43 | 71.67 ± 0.64 | 79.64 ± 0.35 |
| | NC-ALG | 83.02 ± 0.53 | 72.11 ± 0.37 | **80.23 ± 0.81** |
| | AGCL | **83.92 ± 0.54** | **73.10 ± 0.58** | 79.83 ± 0.34 |
| GAT | Random | 80.60 ± 1.42 | 70.94 ± 1.77 | 76.84 ± 3.72 |
| | Standard | 82.06 ± 0.56 | 71.38 ± 0.76 | 77.74 ± 0.84 |
| | AGE | 81.42 ± 0.66 | 70.32 ± 0.74 | 79.50 ± 1.81 |
| | FeatProp | 76.9 ± 1.69 | 59.0 ± 2.81 | 68.3 ± 3.19 |
| | GRAIN | 80.44 ± 0.81 | 70.76 ± 0.37 | 79.67 ± 0.60 |
| | RIM | 82.30 ± 0.70 | 73.08 ± 0.67 | 76.44 ± 1.07 |
| | ALG | 82.92 ± 0.47 | 71.28 ± 0.35 | 78.86 ± 0.53 |
| | GraphPart | 82.59 ± 0.82 | 70.78 ± 0.65 | 77.76 ± 0.61 |
| | NC-ALG | 82.63 ± 0.93 | 71.27 ± 0.47 | 79.22 ± 1.32 |
| | AGCL | **83.68 ± 0.39** | **72.92 ± 0.57** | **79.82 ± 0.50** |
| APPNP | Random | 82.15 ± 0.85 | 72.03 ± 1.07 | 77.84 ± 4.18 |
| | Standard | 82.86 ± 0.28 | 71.07 ± 0.76 | 80.12 ± 0.32 |
| | AGE | 83.68 ± 0.26 | 71.43 ± 0.48 | 80.42 ± 1.18 |
| | FeatProp | 78.1 ± 1.56 | 66.3 ± 1.91 | 75.2 ± 1.32 |
| | GRAIN | 82.27 ± 0.74 | 71.35 ± 0.20 | 80.55 ± 0.36 |
| | RIM | 83.18 ± 0.34 | 74.22 ± 0.37 | 76.29 ± 0.42 |
| | ALG | 84.59 ± 0.19 | 72.17 ± 0.10 | 80.05 ± 0.09 |
| | GraphPart | 82.86 ± 0.28 | 71.21 ± 0.89 | 80.12 ± 0.32 |
| | NC-ALG | 84.66 ± 0.40 | 71.73 ± 0.59 | 80.25 ± 0.30 |
| | AGCL | **84.93 ± 0.42** | **73.53 ± 0.42** | **80.91 ± 0.34** |

Table 2: Classification accuracy (%) on three heterophilic datasets with different training sets (mean accuracy (%) and standard deviation over 5 different runs).

| Methods | Training Data | Actor | Squirrel | roman-empire |
|---|---|---|---|---|
| GCN | Random | 28.47 ± 0.93 | 25.94 ± 1.67 | 16.63 ± 2.12 |
| | AGE | 25.38 ± 0.38 | 23.50 ± 1.32 | 10.15 ± 3.83 |
| | GRAIN | 26.47 ± 0.51 | 25.13 ± 0.31 | 4.17 ± 0.00 |
| | AGCL | **29.14 ± 2.52** | **27.53 ± 1.19** | **20.31 ± 0.96** |
| H2GCN | Random | 31.51 ± 0.68 | 34.68 ± 0.95 | 21.36 ± 0.30 |
| | AGE | 28.71 ± 1.68 | 27.26 ± 2.25 | 18.25 ± 1.92 |
| | GRAIN | 31.63 ± 0.95 | 33.24 ± 0.47 | 12.23 ± 0.24 |
| | AGCL | **32.12 ± 0.39** | **35.81 ± 0.82** | **21.71 ± 0.22** |
| GPRGNN | Random | 28.37 ± 1.41 | 25.55 ± 1.35 | 13.93 ± 0.06 |
| | AGE | 25.67 ± 0.83 | 21.88 ± 1.15 | 7.83 ± 3.51 |
| | GRAIN | 26.61 ± 0.51 | 26.26 ± 0.57 | 7.20 ± 3.94 |
| | AGCL | **28.75 ± 0.66** | **28.45 ± 0.63** | **14.01 ± 0.03** |

This result highlights the efficiency of AGCL. The results on Table 4 demonstrates that our proposed method can be extended to large-scale graphs.

While the homophilic datasets are graphs with high **Homo.** (indicating the proportion of edges connecting nodes with the same label (Zhu et al. (2020))), we also consider heterophilic datasets with low **Homo.**. The prediction accuracies for node classification on three different heterophilic

datasets are reported in Table 2. It can be observed that our proposed AGCL method achieves state-of-the-art or competitive performance on all heterophilic datasets across various GNN models (H2GCN Zhu et al. (2020) and GPRGNN Chien et al. (2021) are specially designed heterophily-based methods). The baseline methods fail to achieve better performance compared to random sampling because they cannot explore and exploit more complex structural information, such as the long-range dependent information in heterophilic datasets. In contrast, AGCL captures global-level graph structural information by directly learning the correlations between labeled and unlabeled data from a global perspective, which provides a significant advantage. We can achieve the similar observations on Penn94 which is a large-scale heterophilic datasets in Table 5.

**Training Efficiency.** Table 3 reports the training time of different graph AL methods on cora, citeseer, and Pubmed. We can observe that AGCL is orders of magnitude faster than some graph AL methods. Specifically, AGCL yields 3x training speedup over AGE on cora, and 2x training speedup over ACL on citeseer. In terms of memory usage, AGCL shows memory consumption of 1034.45 MB and 1496.38 MB on the Cora and Citeseer datasets, respectively. Some methods, such as those based on query heuristics like diversity or density, generally require lower memory usage but tend to incur higher time costs and achieve lower performance. AGCL strikes a balance between memory efficiency and training speed, making it a more scalable solution for various datasets.

Table 3: Efficiency comparison of AGCL and other graph AL competitors w.r.t. training time (s) on NVIDIA A100.

| Method | Cora | Citeseer | Pubmed |
|--------|------|----------|--------|
| AGE | 57.45 | 71.68 | 978.14 |
| ACL | 38.23 | 62.36 | 176.08 |
| GRAIN | 21.55 | 37.81 | 172.73 |
| AGCL | 20.15 | 27.92 | 145.83 |

Table 4: Classification accuracy (%) on ogbn-arxiv dataset with different training sets. OOM denotes out-of-memory.

| Training Data | GCN |
|---------------|-----|
| Random | $63.35 \pm 1.01$ |
| AGE | $63.64 \pm 0.78$ |
| GRAIN | OOM |
| AGCL | $64.48 \pm 0.11$ |

Table 5: Classification accuracy (%) on the Penn94 dataset with different training sets selected by graph active learning methods.

| Method | H2GCN | GPRGNN | LINKX |
|--------|-------|--------|-------|
| Random | $66.63 \pm 0.67$ | $67.29 \pm 0.82$ | $65.26 \pm 1.06$ |
| AGE | $66.83 \pm 0.22$ | $66.75 \pm 0.35$ | $65.49 \pm 0.47$ |
| GRAIN | $66.49 \pm 0.09$ | $67.45 \pm 0.40$ | $65.71 \pm 0.24$ |
| AGCL | $66.91 \pm 0.20$ | $67.89 \pm 0.36$ | $67.93 \pm 0.81$ |

## 6 CONCLUSION

In conclusion, we have presented an Attention-based Graph Coreset Labeling (AGCL) framework for graph labeling. Our approach addresses the limitations of existing graph active learning methods in capturing comprehensive graph structural information by connecting labeled and unlabeled data using an attention architecture. Through AGCL, we effectively identify the most informative unlabeled sample for the labeled pool, gradually expanding the labeled dataset to cover the entire graph representation space. This results in improved sample complexity and diversity, leading to enhanced performance of GNNs across various types of graph data. In theoretical analysis, we clear show that selecting unlabeled data with maximum representation difference from the labeled pool helps reduce the bound radius $\delta$, thereby increasing coverage in the representation space. Empirical evaluations across different graph datasets and image classification tasks demonstrate the effectiveness of AGCL in improving the generalization ability of graph models. While the proposed AGCL can be applied to large-scale datasets by operating on subgraphs or subsets, we aim to further explore alternative methods to address efficiency challenges in large-scale datasets. In the paper, we assume a practical scenario where the dataset does not contain a significant amount of outlier data. If there are many anomalies in the dataset, the proposed method may select points that have different representations from the majority of the data. Therefore, it is necessary to consider excluding these anomalies. In the future, we will further consider this case and to achieve class-balanced coreset.

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

# A APPENDIX

## A.1 COMPLEXITY ANALYSIS AND ALGORITHM OF AGCL

In AGCL, the complexity of the attention-based networks is $\mathcal{O}(LU)$, where $L$ is the number of labeled data, $U$ is the number of unlabeled nodes. For data selection with a greedy searching method, the complexity is $\mathcal{O}(LUm + U log(U))$, where $m$ is the dimension of the low-dimensional embedding.

---

**Algorithm 1** Attention-based Graph Coreset Labeling

---

**Input**: Graph $\mathcal{G} = (\mathcal{V}, \mathcal{E})$, batch budget $b$, labeling budget $\mathcal{B}$.
**Output**: Labeled pool $s$.
1: Initialize labeled pool $s = s^0$ with labels
2: **While** $|s| < \mathcal{B}$ **do**
3:    Get the hidden embedding of all data by a GNN model
4:    Get the node representations based on attention-based networks
5:    **for** batch $i = 0, 1, 2, \ldots, b - 1$ **do**
6:       Select node $u_i$ according to Equation equation 9
7:       $\mathbf{s} = \mathbf{s} \cup \{u_i\}$
8:    **end for**
9: **End while**

---

## A.2 PROOF

**Proof of Proposition 4.1**

*Proof.* We proof Proposition 4.1 based on the following assumptions.

**Assumption 1** (local curvature). *For a representation $\mathbf{h}$, both $\frac{d}{dh}\mathcal{L}(f(h))$ and $\frac{d^2}{d^2h}\mathcal{L}(f(h))$ exist and and are continuous and bounded.*

**Assumption 2** (well trained model). For a given training set $\mathbf{V}_l$, and a well-trained GNN model $\mathbf{M}$, for any $\epsilon > 0$ and $v \in \mathcal{V}_L$, we have $l_M(f(\mathbf{h}_v)) < \epsilon$

Assumption 1 regarding local curvature is a standard technical assumption to make the analysis feasible. Assumption 2 has been formally proved Keriven & Peyré (2019), demonstrating that GNNs can achieve universal approximation power. Under mild conditions and with enough parameters in the model, a model with universal approximation power can achieve zero loss on the training set upon convergence, i.e., the property in Assumption 2 Su et al., with high probability.

Let $\theta_D$ be the set of parameters learnt by the GNN model $\mathcal{M}$ that satisfy properties given in Assumption 2. For a labeled sample $v$ with representation $\mathbf{h}_v$, we have $l_{\theta_D}(f(\mathbf{h}_v)) < \epsilon$ with $\epsilon > 0$, i.e., it achieves the global minimum of the loss function in the embedding space, where $\mathbf{h}_v = \mathcal{M}(v)$, $f(\cdot)$ is the prediction function, and $l(\cdot)$ is the loss function. Here, we assume that both $f(\cdot)$ and $l(\cdot)$ are smooth.

According to Assumption 1, we have $\frac{d}{dh}l(f(\mathbf{h}_v)) = 0$ since it achieves a local minimum. Furthermore, the global minimum also states that $\frac{d^2}{d^2h}l(f(\mathbf{h}_v)) \geq 0$.

Assume there are two embeddings $\mathbf{h}$ and $\mathbf{h}'$, where $\Delta(\mathbf{h}, \mathbf{h}_v) \leq \Delta(\mathbf{h}', \mathbf{h}_v) \leq r_v$, with $\Delta$ measuring the distance in the embedding space and $r_v$ defining the width of the neighborhood around $h_v$ in the embedding space. This indicates that $\mathbf{h}$ is closer to $\mathbf{h}_v$ than $\mathbf{h}'$, we have $\mathbf{h}' = \mathbf{h} + \delta$, $\delta > 0$. Thus we can get:

$$l(f(\mathbf{h}')) = l(f(\mathbf{h} + \delta)) \geq l(f(\mathbf{h})) + \frac{d}{dh}l(f(\mathbf{h}))\|\delta\| \tag{11}$$

As $\frac{d}{dh}l(f(\mathbf{h})) \geq 0$ and $\|\delta\| > 0$, we have $l(f(\mathbf{h})) < l(f(\mathbf{h}'))$. $\square$

Based on Proposition 4.1, we provide the proof for Lemma 4.1.

**Proof of Lemma 4.1** Assume there are two training set $\mathcal{V}_{train}$ and $\mathcal{V}'_{train}$, and test set $\mathcal{V}_{test}$. Based on two training set, we get two trained GNN models $\mathcal{M}(\mathcal{V}_l)$ and $\mathcal{M}(\mathcal{V}'_l)$. If $\sum_{v \in \mathcal{V}_{train}} d(v, \mathcal{V}_{test}) < \sum_{u \in \mathcal{V}'_{train}} d(u, \mathcal{V}_{test})$, thus the covering radius $\delta_{v \in \mathcal{V}_{train}} < \delta_{u \in \mathcal{V}'_{train}}$, we have $\sum_{u \in \mathcal{V}_{test}} l(f_{\mathcal{M}(\mathcal{V}_{train})}(u)) < \sum_{u \in \mathcal{V}_{test}} l(f_{\mathcal{M}(\mathcal{V}'_{train})}(u))$.

*Proof.* Assume there are two training set $\mathcal{V}_{train}$ and $\mathcal{V}'_{train}$, and test set $\mathcal{V}_{test}$. Let $G = (V, E)$ be the input graph with node feature vector $X_v$ for all $v \in V$. Let $\mathcal{M}$ be a given GNN model and $f$ be the prediction function that maps the output of $\mathcal{M}$ to the class representation. The loss function $l$ is $\lambda^l$ Lipschitz continuous for all $y$ bounded by $L$. According to Proposition 4.1, we can obtain zero-error for labeled data and have

$$\sum_{v \in \mathcal{V}_{train}} l\left(f_{\mathcal{M}(\mathcal{V}_{train})}(v)\right) = 0,$$

$$\sum_{v \in \mathcal{V}'_{train}} l\left(f_{\mathcal{M}(\mathcal{V}'_{train})}(v)\right) = 0.$$

Then, we consider the loss on the test set $\mathcal{V}_{test}$ with the model trained on two training sets, which can be written as:

$$\sum_{u \in \mathcal{V}_{test}} \mathcal{L}\left(f_{\mathcal{M}(\mathcal{V}_{train})}(u)\right),$$

$$\sum_{u \in \mathcal{V}_{test}} \mathcal{L}\left(f_{\mathcal{M}(\mathcal{V}'_{train})}(u)\right).$$

From Proposition 4.1, we known that the loss function is monotonically increasing with respect to the embedding distance in $\delta_{\boldsymbol{h}_v}$, where $\boldsymbol{h}_v$ is the hidden representation based on trained model $\mathcal{M}$.

According to Theorem 1 in Sener & Savarese (2018), we know that the loss function is bounded by convering radius $\delta$. Now we extend the similar conclusion to GNN: We have a condition which states that there exists $h_j$ in $\delta$ ball around $h_i$ such that $h_j$ has 0 loss.

$$
\begin{aligned}
E_{y_i \sim \eta(h_i)} \left[l_{\mathcal{M}}\left(\mathcal{G}, y_i; A_{\mathbf{s}}\right)\right] &= \sum_{k \in [C]} p_{y_i \sim \eta_k(h_i)}(y_i = k) l_{\mathcal{M}}\left(\mathcal{G}, k; A_{\mathbf{s}}\right) \\
&\overset{(d)}{\le} \sum_{k \in [C]} p_{y_i \sim \eta_k(h_j)}(y_i = k) l_{\mathcal{M}}\left(\mathcal{G}, k; A_{\mathbf{s}}\right) \\
&\quad + \sum_{k \in [C]} \left|\eta_k\left(h_i\right) - \eta_k\left(h_j\right)\right| l_{\mathcal{M}}\left(\mathcal{G}, k; A_{\mathbf{s}}\right) \\
&\overset{(e)}{\le} \sum_{k \in [C]} p_{y_i \sim \eta_k(h_j)}(y_i = k) l_{\mathcal{M}}\left(\mathcal{G}, k; A_{\mathbf{s}}\right) + \delta \lambda^\eta L C
\end{aligned}
\tag{12}
$$

We use the Claim in Berlind & Urner (2015), i.e., fix $p, p' \in [0, 1]$ and $y' \in [0, 1]$, then, $p_{y \sim p}\left(y \ne y'\right) \le p_{y \sim p'}\left(y \ne y'\right) + |p - p'|$ to achieve $(d)$, and use Lipschitz property of regression function and bound of loss in $(e)$. Then, we further bound

$$
\begin{aligned}
\sum_{k \in [C]} p_{y_i \sim \eta_k(h_j)}\left(y_i = k\right) l_{\mathcal{M}}\left(\mathcal{G}, k; A_{\mathbf{s}}\right) &= \sum_{k \in [C]} p_{y_i \sim \eta_k(h_j)}\left(y_i = k\right) \left[l\left(h_i, k; A_{\mathbf{s}}\right) - l\left(h_j, k; A_{\mathbf{s}}\right)\right] \\
&\quad + \sum_{k \in [C]} p_{y_i \sim \eta_k(h_j)}\left(y_i = k\right) l\left(h_j, k; A_{\mathbf{s}}\right) \\
&\le \delta \lambda^l
\end{aligned}
\tag{13}
$$

where last step is coming from the fact that the trained classifier assumed to have 0 loss over training data. Here, $l_{\mathcal{M}}(\mathcal{G}, y_i; A_{\mathbf{s}}) = l(h_i, y_i; A_{\mathbf{s}})$, as $h_i$ is the low-dimensional embedding of $x_i$ by GNN $\mathcal{M}$. Then, we can get

$$E_{y_i \sim \eta(h_i)}\left[l_{\mathcal{M}}(\mathcal{G}, k; A_{\mathbf{s}})\right] \leq \delta\left(\lambda^l + \lambda^\eta LC\right). \tag{14}$$

We further use Hoeffding's inequality Hoeffding (1994) and finally obtain

$$\left|\frac{1}{N}\sum_{i\in[N]} l_{\mathcal{M}}(\mathcal{G}, y_i; A_{\mathbf{s}}) - \frac{1}{|\mathbf{s}|}\sum_{j\in\mathbf{s}} l_{\mathcal{M}}(\mathcal{G}, y_j; A_{\mathbf{s}})\right| \leq \delta\left(\lambda^l + \lambda^\eta LC\right) + L\sqrt{\frac{\log(1/\gamma)}{2N}} \tag{15}$$

with probability at least $1 - \gamma$.

Thus, while $\sum_{v\in\mathcal{V}_{train}} d(v, \mathcal{V}_{test}) < \sum_{u\in\mathcal{V}'_{train}} d(u, \mathcal{V}_{test})$, we have he covering radius $\delta_{v\in\mathcal{V}_{train}} < \delta_{u\in\mathcal{V}'_{train}}$. The smaller covering radius means the smaller loss for the whole samples, thus, we have $\sum_{u\in\mathcal{V}_{test}} l(f_{\mathcal{M}(\mathcal{V}_{train})}(u)) < \sum_{u\in\mathcal{V}_{test}} l(f_{\mathcal{M}(\mathcal{V}'_{train})}(u))$.

□

**Proof of Theorem 4.1**

*Proof.* Given the whole sample $\mathcal{V}$ drawn from $\mathcal{G}$, let $\mathcal{V}_l$ represent the labeled pool consisting of points with labels, and let $\mathcal{V}_u$ denote the set of unlabeled data. $\exists s \in \mathcal{V}_u : \forall v \in \mathcal{V}_l, d(s, v) < d(k, v)$ with $k \in \mathcal{V}_u \setminus \{s\}$, thus we have $\sum_{v\in\mathcal{V}_l\cup\{s\}} d(v, \mathcal{V}) < \sum_{u\in\mathcal{V}_l\cup\{k\}} d(u, \mathcal{V})$ where $\mathcal{V}$ denote the whole graph data. In other word, we have $\sum_{v\in\mathcal{V}_l\cup\{s\}} d(v, \mathcal{V}_{test}) < \sum_{u\in\mathcal{V}_l\cup\{k\}} d(u, \mathcal{V}_{test})$ for test set $\mathcal{V}_{test}$. Acoording to Lemma 4.1, we can get that $\delta_{v\in\mathcal{V}_l\cup\{s\}} < \delta_{v\in\mathcal{V}_l\cup\{k\}} < \delta_{u\in\mathcal{V}_l}$. Thus, we have $\sum_{i\in\mathcal{V}} l(f_{\mathcal{M}(\mathcal{V}_l\cup\{s\})}(i)) < \sum_{i\in\mathcal{V}} l(f_{\mathcal{M}(\mathcal{V}_l\cup\{k\})}(i)) < \sum_{i\in\mathcal{V}} l(f_{\mathcal{M}(\mathcal{V}_l)}(i))$. □

A.3 EXPERIMENTAL PART

Table 6: Implementation details

| Model | Dataset | Hyper-parameter | | | |
|---|---|---|---|---|---|
| | | Epochs | Learning Rate | Weight Decay | Hidden Units |
| GCN, GAT, APPNP | Cora, Citeseer, Pubmed | 200 | 1e-2 | 5e-4 | 64 |
| | ogbn-arxiv | 300 | 1e-2 | 0 | 64 |

**Implementation details for image classification.** ResNet-18 [15] is the favourite choice as learner due to its relatively higher accuracy and better training stability. During training the learner, we set a batch size of 64. We use Stochastic Gradient Descent (SGD) with a weight decay $5e-4$ and a momentum of 0.9. At every selection stage, we train the model for 200 epochs. We set the initial learning rate of 0.1 and decrease it by the factor of 10 after 160 epochs. We use the same set of hyper-parameters in all the experiments. For quantitative evaluation, we report the mean average accuracy of 5 trials on the test sets.

A.4 DATASET STATISTIC.

Table 7: Statistics of graph benchmark datasets.

| | Cora | Citeseer | Pubmed | ogbn-arxiv | Actor | roman-empire | Squirrel |
|---|---|---|---|---|---|---|---|
| **# Nodes** | 2,708 | 3,327 | 19,717 | 169,343 | 7,600 | 22,662 | 24,492 |
| **# Edges** | 5,429 | 4,732 | 44,338 | 1,166,343 | 26,752 | 32,927 | 93,050 |
| **Class** | 7 | 6 | 3 | 40 | 5 | 18 | 5 |

Table 8: The performance of GraphSAGE on three citation datasets with different training sets.

| Methods | Training data | Cora | Citeseer | Pubmed |
|---|---|---|---|---|
| | GRAIN | $81.55 \pm 0.50$ | $71.06 \pm 0.44$ | $\mathbf{79.23 \pm 0.30}$ |
| GraphSAGE | GraphPart | $81.27 \pm 0.43$ | $70.42 \pm 0.58$ | $77.29 \pm 0.35$ |
| | AGCL | $\mathbf{82.56 \pm 0.39}$ | $\mathbf{72.64 \pm 1.11}$ | $79.14 \pm 0.33$ |

## A.5 THE RESULTS ON GRAPHSAGE.

We have conducted additional experiments to evaluate AGCL with GraphSAGE (Hamilton et al. (2017)). The results, shown in the Table 8, indicate that AGCL performs consistently well across different datasets. Specifically, AGCL outperforms other methods when using GraphSAGE, further demonstrating its versatility and effectiveness in core data selection, irrespective of the underlying GNN model.

## A.6 MORE RESULTS.

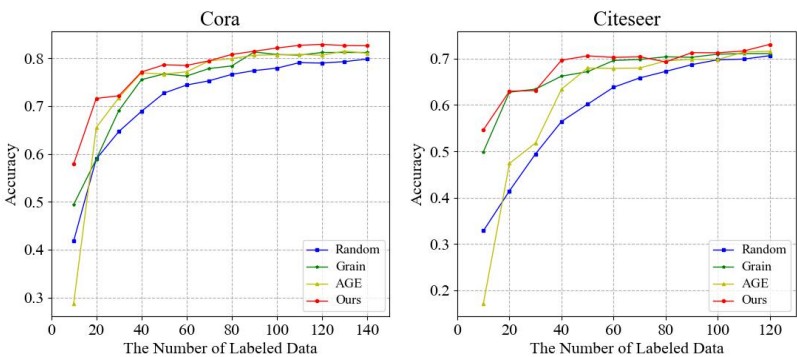

Figure 2: The test accuracy across different labeling budgets for GCN model training.

## A.7 RESULTS ON IMAGE CLASSIFICATION

To demonstrate the generalization ability of our proposed method, we report the performance comparison of AGCL with six existing methods on CIFAR-10 and FashionMNIST datasets in Figure 3. Our proposed attention-based graph coreset labeling method can achieve the comparative or even better performance compared to some CNN baselines. Especially, after selecting 4000 labeled examples, the AGCL achieves highest performances with 82.23% and 89.97% on CIFAR-10 and FashionMNIST, respectively.

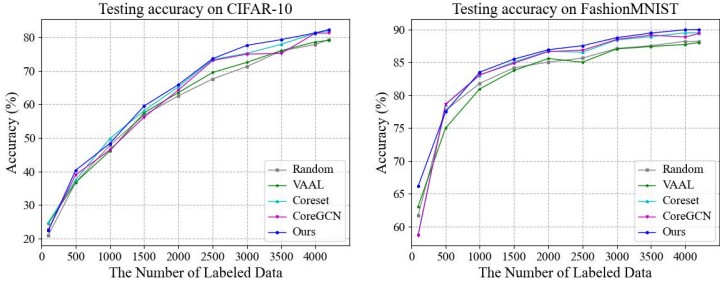

Figure 3: The comparison of several active learning methods on CIFAR-10 and FashionMNIST. The accuracy is averaged over 5 runs.

## A.8 ABLATION STUDY

Positional encoding plays a crucial role in the self-attention framework. Based on GCNs, we test its impact on the proposed AGCL framework by comparing two common positional encoding methods: Laplacian-based (lpe) and random walk positional encoding (rwpe) against AGCL without any positional encoding (w/o pe).

As shown in Table 9, we observe that the GCN achieves superior performance when trained on data selected by AGCL with positional encoding, compared to AGCL without positional encoding. The difference in performance is minimal with either Laplacian-based or random walk positional encoding methods across all three datasets.

Table 9: Ablation study on positional encoding in AGCL.

| Methods | Training Data | Cora | Citeseer | pubmed |
|---------|---------------|------|----------|--------|
|         | AGCL wo/ pe   | $82.59 \pm 0.37$ | $71.93 \pm 0.64$ | $79.53 \pm 0.55$ |
| GCN     | AGCL (lpe)    | $83.92 \pm 0.54$ | $73.16 \pm 0.46$ | $79.10 \pm 0.90$ |
|         | AGCL (rwpe)   | $82.89 \pm 0.38$ | $73.10 \pm 0.58$ | $79.83 \pm 0.34$ |

## A.9 ROBUSTNESS ANALYSIS OF AGCL WITH NOISY DATA

We further demonstrate the robustness of the proposed method on noisy data. Specifically, we simulate noisy data by randomly removing a certain percentage (10%) of the graph node features.

As shown in Table 10, we observe that AGCL continues to achieve strong performance, especially on the Cora and Citeseer datasets, in the presence of noise. This demonstrates the method's robustness and its ability to perform well in settings that might better resemble real-world, noisy data scenarios.

Table 10: Classification accuracy (%) on three citation datasets with different training sets (noise and clear).

| Methods | Training data | Cora | Citeseer | Pubmed |
|---------|---------------|------|----------|--------|
| GCN     | noise         | $82.58 \pm 0.40$ | $72.04 \pm 0.19$ | $76.08 \pm 0.46$ |
|         | clear         | $83.92 \pm 0.54$ | $73.10 \pm 0.58$ | $79.83 \pm 0.34$ |