# OpenReview forum: "Attention-based Graph Coreset Labeling for Active Learning"
_ICLR.cc/2025/Conference — Submitted to ICLR 2025_

### Official Review · Reviewer_akoo · 2024-10-27

**Soundness:** 2
**Presentation:** 2
**Contribution:** 2
**Rating:** 5
**Confidence:** 4

**Summary:**

This paper proposes an Attention-based Graph Coreset Labeling (AGCL) framework for active learning in graph neural networks. By leveraging attention mechanisms, the proposed AGCL aims to identify the most representative nodes for labeling. It addresses the challenge of selecting informative nodes from a global perspective by focusing on representation diversity between labeled and unlabeled nodess. The experiments demonstrate that AGCL effectively enhances model performance with fewer labeled nodes, which outperforms existing methods across various datasets.

**Strengths:**

AGCL improves label efficiency by selecting nodes that maximize representation diversity, which allows the model to generalize better with fewer labeled samples.

Experiments show that AGCL outperforms existing methods across multiple GNN architectures and datasets

**Weaknesses:**

In the introduction, the authors fail to adequately explain why traditional active learning methods developed for CNNs cannot be directly applied to GNNs, which might diminish the novelty and motivation for the proposed method.

The authors aim to select "truly informative data points" in graph-structured data but do not provide a clear definition or criteria for what makes a data point "informative." A more precise definition of informativeness, with concrete criteria or theoretical support, would be helpful.

The proposed method uses attention coefficients to capture global influence between labeled and unlabeled nodes. However, I question the reliability of these coefficients, given that initial labels may not provide sufficient supervision for accurate training. Could the authors clarify how they ensure the attention coefficients' accuracy under limited labeled data?

The comparison methods used in the paper are somewhat outdated and may not sufficiently demonstrate the advantages of the proposed algorithm. Incorporating more recent state-of-the-art methods would strengthen the evaluation.

**Questions:**

How do the authors ensure that attention coefficients are reliable given the limited labeled data available initially?

Could the authors elaborate on why traditional CNN-based active learning methods are not suitable for GNNs?

---

> ### Author Response · Authors · 2024-11-23
>
> Dear Reviewer,
>
> We would like to express our sincere gratitude for the time and effort you have dedicated to reviewing our work. Below, we provide a detailed response to your comments.
>
> **Q1.**
> In the introduction, the authors fail to adequately explain why traditional active learning methods developed for CNNs cannot be directly applied to GNNs, which might diminish the novelty and motivation for the proposed method.
>
> **A1.**
> We appreciate your feedback on the introduction. To address your concern, we will revise the introduction to provide a more comprehensive explanation of why traditional active learning methods developed for CNNs are not directly applicable to GNNs.
>
> Traditional active learning (AL) strategies are designed for models that operate on independent and identically distributed (i.i.d.) data. As a result, these methods fail to capture both the graph structure and node features, leading to suboptimal performance when applied to GNNs [1, 2]. Therefore, it is essential to develop graph-based active learning methods that are capable of selecting the most informative samples by considering both node features and the graph structure.
>
> [1] Li Gao, et al. Active Discriminative Network Representation Learning. IJCAI 2018.
>
> [2] Kaushalya Madhawa and Tsuyoshi Murata. 2020. Active Learning for Node Classification: An Evaluation. Entropy 22, 10 (2020), 1164.
>
> **Q2.**
> The authors aim to select "truly informative data points" in graph-structured data but do not provide a clear definition or criteria for what makes a data point "informative." A more precise definition of informativeness, with concrete criteria or theoretical support, would be helpful.
>
> **A2.**
> The theoretical analysis in the paper aims to support the coreset selection strategy, which focuses on selecting the most informative data that minimizes the loss in the representation space. An informative sample can be defined as follows:
>
> **Definition. (Informative Sample)**
> Given the whole sample $\mathcal{V}$ drawn from $\mathcal{G}$, labeled pool $\mathcal{V}_l$ and
>
> unlabeled pool $\mathcal{V}_u$.
>
> We consider the data point $s$ is the most informative sample in $\mathcal{V_u}$ based on the existing labeled pool $\mathcal{V_l}$,
>
> if
> $\sum_{i \in \mathcal{V}} l (f_{\mathcal{M}(\mathcal{V}_l \cup \{s\} )} (i)) $
>
> $< \sum_{i \in \mathcal{V}} l (f_{\mathcal{M}(\mathcal{V}_l \cup \{k\}}(i))$
>
> $ < \sum_{i \in \mathcal{V}} l (f_{\mathcal{M}(\mathcal{V}_l }(i))$
>
> when training on GNN model $\mathcal{M}$. Where $l(\cdot)$ is the loss function, $k \in \mathcal{V}_u \setminus \{s\}$. This condition ensures that the model trained with the labeled pool $\mathcal{V}_l$ augmented by the most informative sample $s$ results in the smallest loss, compared to any other unlabeled sample $k \in \mathcal{V}_u$.
>
> **Q3.**
> The proposed method uses attention coefficients to capture global influence between labeled and unlabeled nodes. However, I question the reliability of these coefficients, given that initial labels may not provide sufficient supervision for accurate training. Could the authors clarify how they ensure the attention coefficients' accuracy under limited labeled data?
>
> **A3.**
> Thank you for pointing out this question. Under the constraint of limited labeled data, achieving reliable attention coefficients is indeed challenging. To address this issue, AGCL introduces new method to select the initial labeled pool through a systematic search.
>
> **Structure and Feature-based Initial Pool Selection.**
>
> Considering both the features and graph structure, we propagate features among nodes with the layer-wise propagation rule:
> \begin{equation}
> {\boldsymbol{H}}^{(l+1)}=\hat{\mathbf{A}} {\boldsymbol{H}}^{(l)},
> \end{equation}
> where $\hat{\textbf{A}}=\hat{\mathbf{D}}^{-1 / 2}(\textbf{A}+\textbf{I}) \hat{\mathbf{D}}^{-1 / 2}$ is a symmetric normalized adjacency matrix, $\mathbf{I}$ is the identity matrix, $\hat{\mathbf{D}}$ is the corresponding degree matrix of $\mathbf{{A}}+\mathbf{I}$, and  $\boldsymbol{H}^{(l)}$ is the hidden node representation in $l^{th}$ layer with ${\boldsymbol{H}}^{(0)}= \mathbf{{X}}$.  After $k$ iterations of aggregation, the representation of a node $h_i^k$ captures the structural information within its $k$-hop neighborhood. Then, we select $k$ nodes into the initial pool using the k-medoids method.
>
>
> **Hyperparameter Search for Initial Labels:**
> For the number of initial labels $|s^0|$, we perform a hyperparameter search for each dataset to further achieve reliable attention coefficients.
> In our experiments, we observed that setting $S^0 = 0.3*B$ is effective for commonly used datasets, enabling the selection of a representative coreset from the unlabeled data.

---

> > ### Author Response · Authors · 2024-11-23
> >
> > Thanks for your time! Here is the rest part.
> >
> > **Q4.**
> > The comparison methods used in the paper are somewhat outdated and may not sufficiently demonstrate the advantages of the proposed algorithm. Incorporating more recent state-of-the-art methods would strengthen the evaluation.
> >
> > **A4.**
> > We have further evaluated the effectiveness of the proposed AGCL by comparing it with NC-ALG [1], a more recent graph-based active learning method. NC-ALG employs a novel metric to quantify influence reliability and leverages an effective influence maximization objective for node selection.
> >
> > In our experiments, as shown in the following table, AGCL consistently outperforms NC-ALG across most of the datasets, highlighting its superior ability to select truly informative nodes for active learning.
> >
> > **Table: Classification accuracy (\%) on three citation datasets with different training sets.**
> >
> > | **Methods**      | **Training data** | **Cora**         | **Citeseer**     | **Pubmed**       |
> > |------------------|-------------------|------------------|------------------|------------------|
> > | **GCN**          | NC-ALG            | 83.02 ± 0.53     | 72.11 ± 0.37     | **80.23** ± **0.81** |
> > |                  | AGCL              | **83.92** ± **0.54** | **73.10** ± **0.58** | 79.83 ± 0.34     |
> > | **GAT**          | NC-ALG            | 82.63 ± 0.93     | 71.27 ± 0.47     | 79.22 ± 1.32     |
> > |                  | AGCL              | **83.92** ± **0.54** | **73.10** ± **0.58** | **79.83** ± **0.34** |
> > | **APPNP**        | NC-ALG            | 84.66 ± 0.40     | 71.73 ± 0.59     | 80.25 ± 0.30     |
> > |                  | AGCL              | **84.93** ± **0.42** | **73.53** ± **0.42** | **80.91** ± **0.34** |
> >
> > [1] Zhang, Wentao, et al. "NC-ALG: Graph-Based Active Learning Under Noisy Crowd." 2024 IEEE 40th International Conference on Data Engineering (ICDE). IEEE, 2024.

---

> > > ### Author Response · Authors · 2024-11-25
> > >
> > > Dear Reviewer,
> > >
> > > Thank you for taking the time and effort to review our paper. We have thoughtfully addressed each question you raised and hope our responses alleviate your concerns.
> > >
> > > In the updated manuscript (highlighted in blue), we have incorporated your suggestions, including a more detailed explanation of why CNNs cannot be directly applied to GNNs, connecting "truly informative data" to theoretical support, and providing additional experimental results on recent methods. We hope these updates help clarify your uncertainties.
> > >
> > > As the discussion deadline approaches, we would greatly appreciate any further suggestions you might have for improving our work. Your feedback is invaluable, and we welcome your guidance.
> > >
> > > Considering that all comments have been duly addressed, we humbly ask you to reconsider the rating. Thank you once again for your insightful review and consideration.
> > >
> > > Best regards,
> > >
> > > The Authors

---

> > > > ### Comment · Reviewer_akoo · 2024-12-03
> > > > **To authors**
> > > >
> > > > Thank you for your response. The score has been increased from 3 to 5. However, concerns remain regarding the reliability of the attention coefficients in capturing global influence, and the inclusion of more recent methods is insufficient, which prevents the score from being raised further.

---

> ### Author Response · Authors · 2024-12-03
>
> Dear Reviewer,
>
> We sincerely appreciate your acknowledgment of our efforts to address your initial concerns. We would like to further clarify and expand on the points you raised.
>
> **1. The reliability of the attention coefficients.**
>
> In our paper, AGCL introduces a Structure and Feature-based Initial Pool Selection method, accompanied by a hyperparameter search strategy, to select initial labels. This approach is grounded in the principle that less but high-quality data can lead to  reliable training—an idea encapsulated in the adage "less is more".
>
> This principle has been validated across various domains and tasks. For instance:
>
> - In large language models (LLM), fewer high-quality data points have been shown to suffice for effective pretraining [1, 2] and fine-tuning [3].
>
> - In graph-based domains, studies [4] have demonstrated that a smaller subset of graph data can yield excellent performance for node classification tasks.
>
> In our experiments, AGCL consistently achieves superior performance compared to other graph active learning methods. As shown in Figure 2 (Appendix A.6), AGCL rapidly boosts accuracy in the early stages of training and consistently outperforms baselines as the number of labeled nodes increases.
>
> These results suggest that the attention coefficients in AGCL are indeed reliable for selecting informative samples. AGCL benefits from combining heuristic-based initial label selection with a learned measure for coreset selection, ensuring effective and robust performance.
>
> **2. Recent methods.**
>
> In the experiments, we compare the proposed AGCL with nine baselines, including recent and widely adopted graph active learning (AL) methods in the graph domain. The effectiveness of AGCL is demonstrated through extensive experiments across diverse data types (homophily and heterophily) and data scales (same-scale and large-scale). Furthermore, we evaluate AGCL against four baselines in the image domain, further highlighting its evident advantages in coreset selection.
>
> We have actively sought to compare AGCL with more recent works, such as Bayesian uncertainty sampling [5] and DOCTOR, which applies the Expected Model Change Maximization (EMCM) principle to GNNs [6]. However, we could not obtain official code for these works, limiting our ability to reproduce their results in the time available. Additionally, these methods still rely on query heuristics, which 1) lack a direct correlation with the expected prediction performance on unlabeled nodes in the final task, and 2) mainly focus on the local graph structure (we will add more discussion in the Related Work section).
>
> If you are aware of additional recent methods related to our study, we would greatly appreciate your recommendations. We are eager to explore further comparisons and discussions to strengthen our work.
>
> Once again, we thank you for your valuable feedback and hope this response adequately addresses your concerns.
>
> [1] Marion, Max, et al. "When less is more: Investigating data pruning for pretraining llms at scale." arXiv preprint arXiv:2309.04564 (2023).
>
> [2] Gunasekar, Suriya, et al. "Textbooks are all you need." arXiv preprint arXiv:2306.11644 (2023).
>
> [3] Zhou, Chunting, et al. "Lima: Less is more for alignment." Advances in Neural Information Processing Systems 36 (2024).
>
> [4] Fu, Sichao, et al. "Finding core labels for maximizing generalization of graph neural networks." Neural Networks 180 (2024): 106635.
>
> [5] Fuchsgruber, Dominik, et al. "Uncertainty for Active Learning on Graphs." arXiv preprint arXiv:2405.01462 (2024).
>
> [6] Song, Zixing, et al. "No change, no gain: empowering graph neural networks with expected model change maximization for active learning." Advances in Neural Information Processing Systems 36 (2024).

---

### Official Review · Reviewer_T82Q · 2024-10-29

**Soundness:** 2
**Presentation:** 3
**Contribution:** 2
**Rating:** 6
**Confidence:** 3

**Summary:**

The paper "Attention-based Graph Correlation Distillation with Proxy Node" introduces a new framework called Attention-based Graph Coreset Labeling (AGCL) to improve graph neural networks (GNNs) through active learning. The method leverages attention mechanisms to explicitly model correlations between labeled and unlabeled data in graph structures. By selecting unlabeled data with the greatest difference in representation from the current labeled set, AGCL enhances the coverage and diversity of the labeled pool, ultimately reducing prediction loss. The framework is demonstrated to work across various GNN architectures and graph data types (homophilic and heterophilic), and its flexibility allows it to be applied to image classification tasks as well. The paper presents both theoretical proofs and experimental results that show AGCL's superiority over existing graph active learning methods, offering improved accuracy and efficiency for semi-supervised node classification tasks.

**Strengths:**

1. The paper introduces a new attention-based approach that effectively models correlations between labeled and unlabeled data in graph structures. This novel framework enhances graph neural networks (GNNs) by selecting data that maximizes label diversity and reduces prediction loss, providing an innovative solution to active learning in GNNs.

2. The method demonstrates its flexibility by being applicable to both homophilic and heterophilic graphs, as well as image classification tasks. This versatility allows the framework to be broadly useful across various GNN architectures and data types, showing its potential for wide-scale adoption.

3. The paper backs its approach with solid theoretical foundations and extensive experiments, proving the effectiveness of the framework. The empirical results demonstrate superior performance compared to existing methods, highlighting the model's ability to improve accuracy and efficiency in semi-supervised learning.

**Weaknesses:**

1. While the paper demonstrates the method's effectiveness on benchmark datasets, it may not fully address the challenges posed by more complex real-world graphs, which often involve dynamic, evolving structures or noisy data. More diverse, real-world testing would be needed to validate its robustness in various practical settings.

2. While the theoretical analysis in the paper aims to support the coreset selection strategy by minimizing the loss in the representation space, it doesn't fully explore how the attention mechanism or the architecture of the proposed Attention-based Graph Coreset Labeling (AGCL) model specifically affects these theoretical guarantees.

**Questions:**

see weaknesses

---

> ### Author Response · Authors · 2024-11-23
>
> Dear reviewer,
>
> We deeply appreciate the time and effort you have dedicated to reviewing our work. Below, we address each of the concerns and questions you raised.
>
> **Q1.**
> While the paper demonstrates the method's effectiveness on benchmark datasets, it may not fully address the challenges posed by more complex real-world graphs, which often involve dynamic, evolving structures or noisy data. More diverse, real-world testing would be needed to validate its robustness in various practical settings.
>
> **A1.**
> We further demonstrate the robustness of the proposed method on noisy data. Specifically, we simulate noisy data by randomly removing a certain percentage (10\%) of the graph node features.
>
> As shown in the table below, we observe that AGCL continues to select informative data to achieve strong performance, especially on the Cora and Citeseer datasets, even in the presence of noise. This demonstrates the method's robustness and its ability to perform well in settings that might better resemble real-world, noisy data scenarios.
>
> **Table: The performance of GCN on three citation datasets with different training sets (noise and clear).**
>
> | **Methods**      | **Training data** | **Cora**         | **Citeseer**     | **Pubmed**       |
> |------------------|-------------------|------------------|------------------|------------------|
> | **GCN**          | noise             | 82.58 ± 0.40     | 72.04 ± 0.19     | 76.08 ± 0.46     |
> |                  | clear             | 83.92 ± 0.54.   |    73.10 ± 0.58   |  79.83 ± 0.34 |
>
> **Q2.** While the theoretical analysis in the paper aims to support the coreset selection strategy by minimizing the loss in the representation space, it doesn't fully explore how the attention mechanism or the architecture of the proposed Attention-based Graph Coreset Labeling (AGCL) model specifically affects these theoretical guarantees.
>
> **A2.**
>
> **Exploring the Intuition Behind AGCL: A Theoretical and Methodological Analysis.**
>
> Theoretically, Proposition 4.1 shows that for two testing samples, the one with a smaller distance to the training sample has better prediction performance.
>
> In Lemma 4.1, we extend Proposition 4.1 to the entire dataset and demonstrate that, for two training/labeled sets, a model trained on the set with closer distances to the remaining/testing/unlabeled data—indicating a smaller covering radius—exhibits better performance.
>
> **Lemma 4.2 shows the connection between coreset analysis and the attention-based score, specifically, demonstrating the benefit of selecting the coreset based on an explicit metric (representation scores) between the labeled and unlabeled pools in improving expected prediction accuracy.**
>
> Building on Lemma 4.1 and Lemma 4.2, Theorem 4.1 states that in data selection, we need to explore samples in the unlabeled pool that have the maximum representation difference from the existing labeled pool in the well-defined representation space.
>
> According to Theorem 4.1, **1)** we need to explicitly show the relationship/distance/influence of unlabeled data and labeled data through a measure that relates to final prediction performance, and **2)** the measure function needs to capture both feature and complex graph structural information from a global perspective.
>
> Based on our theoretical ananlysis, we propose AGCL, a method that directly explores the correlations between labeled and unlabeled data in the embedding space using an attention-based message-passing strategy. Aside from considering the local structure of the graph, attention-based networks also incorporate global structural information.
> By learning the influence between one node in the labeled pool and another in the unlabeled pool and mapping it to the attention matrix, we can intuitively select nodes according to attention scores from a global view.

---

> > ### Author Response · Authors · 2024-11-25
> >
> > Dear Reviewer,
> >
> > Thank you for taking the time and effort to review our paper. In the updated manuscript (highlighted in blue), we have incorporated additional analysis on the robustness of AGCL, supported by experimental results.
> >
> > As the discussion deadline approaches, we would greatly appreciate any further suggestions you might have for improving our work. Your feedback is invaluable, and we warmly welcome your guidance.
> >
> > Thank you once again for your thoughtful review and consideration.
> >
> > Best regards,
> >
> > The Authors

---

> > > ### Comment · Reviewer_T82Q · 2024-11-25
> > > **To authors**
> > >
> > > Thanks for your careful responses. According to the comments of other reviewers, I will keep my score and I think the paper could be better.

---

### Official Review · Reviewer_3m97 · 2024-11-03

**Soundness:** 3
**Presentation:** 3
**Contribution:** 3
**Rating:** 6
**Confidence:** 4

**Summary:**

The paper is about GNN-based Active Learning, which is an interesting topic. The authors propose an Attention-based Graph Coreset Labeling framework, which can gradually discover core data to be labeled from a global view so as to obtain a training dataset that can efficiently depict the whole graph space and maximize the performance of GNNs. The paper is well written and well organized. However, there are several concerns in the current version of the paper that addressing them will increase the quality of this paper.

**Strengths:**

1 This paper focuses on the important issues that do face the GNN community.

2 The paper is clearly structured and easy to understand.

**Weaknesses:**

1 In the abstract, authors should consider introducing more background information for their research so that readers outside the field can better understand it.

2 Figure 1 can be further improved. Currently, we cannot see a clear model structure and the information that can be expressed is limited.

3 Do the authors discuss the computational complexity of the proposed strategy in detail (not just a brief discussion) and analyze the time and memory consumption of the model in the experiments?

4 The datasets used by the authors are limited in size, and the only relatively large dataset has only 160,000+ nodes. This does not require experiments, but I hope the author can further explain what problems the proposed strategy will encounter when facing large-scale datasets, or what benefits it will bring?

**Questions:**

As above weakness 4.

---

> ### Author Response · Authors · 2024-11-23
>
> Dear Reviewer,
>
> Thank you for your constructive comments and suggestions. They have been invaluable in helping us enhance the quality and clarity of our paper. Please find our point-by-point responses to your concerns below.
>
> **Q1.**
>  In the abstract, authors should consider introducing more background information for their research so that readers outside the field can better understand it.
>
> **A1.**
> Thank you for your valuable suggestion. We have revised the abstract and the introduction to include additional background information, which will help readers outside the field better understand the context and motivation behind our research.
>
> **Q2.**
>  Figure 1 can be further improved. Currently, we cannot see a clear model structure and the information that can be expressed is limited.
>
> **A2.**
> Thank you for your feedback on Figure 1. We have made adjustments to the figure to improve its clarity and enhance the model structure representation. We will include the updated version in the revised manuscript.
>
> **Q3.**
> Do the authors discuss the computational complexity of the proposed strategy in detail (not just a brief discussion) and analyze the time and memory consumption of the model in the experiments?
>
> **A3.**
> Table 4 in the paper reports the training time of various graph active learning (AL) methods on the Cora, Citeseer, and Pubmed datasets. From the results, we observe that AGCL achieves significantly faster training times compared to several existing graph AL methods, demonstrating its computational efficiency.
>
> Specifically, AGCL delivers a 3x speedup in training time over AGE on the Cora dataset, and a 2x speedup compared to ACL on the Citeseer dataset. This improvement can be attributed to the streamlined node selection process in AGCL, which effectively reduces the computational overhead associated with more complex selection strategies.
>
> In terms of memory usage, AGCL shows memory consumption of 1034.45 MB and 1496.38 MB on the Cora and Citeseer datasets, respectively. Some methods, such as those based on query heuristics like diversity or density, generally require lower memory usage but tend to incur higher time costs and achieve lower performance. AGCL strikes a balance between memory efficiency and training speed, making it a more scalable solution for various datasets.
>
> **Q4.**
> The datasets used by the authors are limited in size, and the only relatively large dataset has only 160,000+ nodes. This does not require experiments, but I hope the author can further explain what problems the proposed strategy will encounter when facing large-scale datasets, or what benefits it will bring?
>
> **A4.**
> For large-scale datasets with a large number of graph nodes, connecting each node in the labeled pool to nodes in the unlabeled pool can lead to high memory and computational costs, especially when performing operations like attention mechanisms. The challenge arises from the fact that the complexity of such operations scales with the size of the graph.
>
> To address these computational challenges in large-scale datasets, we propose an approach where the dataset is split into subgraphs or subsets. By partitioning the dataset, we can apply the attention-based core set labeling method within each subset, which allows for more efficient data selection while managing resource consumption. This method helps us handle large-scale datasets without overwhelming computational resources.
>
> Additionally, in Appendix A.6, we demonstrate the application of our selection method on randomly selected subsets of unlabeled images for image classification tasks, further showcasing the scalability of our approach.

---

> > ### Author Response · Authors · 2024-11-25
> >
> > Dear Reviewer,
> >
> > Thank you very much for your time and effort in reviewing our paper. Based on your feedback, we have made several updates to the manuscript (highlighted in blue), including:
> >
> > Adding more background information in the abstract to improve clarity and understanding.
> >
> > Updating Figure 1 to enhance visual comprehension.
> >
> > Providing additional analysis on the efficiency of AGCL.
> >
> > Including further analysis of AGCL's applicability to large-scale datasets in the Conclusion.
> >
> > As the discussion deadline approaches, we kindly inquire if you have any additional suggestions for improving our manuscript. Your input is highly valued, and we would greatly appreciate your guidance.
> >
> > Thank you again for your thoughtful review and consideration.
> >
> > Best regards,
> >
> > The Authors

---

> > > ### Comment · Reviewer_3m97 · 2024-11-26
> > >
> > > Thanks to the authors for the reply, I think this paper has merit and I will maintain my positive score.

---

### Official Review · Reviewer_Shvb · 2024-11-03

**Soundness:** 2
**Presentation:** 2
**Contribution:** 2
**Rating:** 5
**Confidence:** 3

**Summary:**

The paper discusses a new active learning algorithm for graph data. Instead of implicit or heuristic based approach such as entropy or graph-based information to select labeled node from unlabeled set, the paper proposed to use attention mechanism to obtain correlation between labeled sets and unlabeled sets and select the labeled nodes based on the obtained correlation score. Based on the concept of coreset and its theoretical basis established in CNN's domain, the author attempts to build a similar case for learned graph embeddings and provides a theoretical foundation for the experimental success of the proposed method. The author conducted experiments on different scales of graph datasets and different homo scores and empirically show the advantage of such proposed method. Additionally, the paper shows advantage of the proposed active learning algorithm on the image classification problem as well.

**Strengths:**

1. The paper provides a comprehensive experiments over different scales of graph datasets as well as different homo levels of graph datasets. Additionally, the paper provides results over different models, demonstrating empirically the proposed methods show some advantages compared to existing active learning approaches on the graph dataset.
2. The paper attempts to establish theoretical basis for the empirical success of the proposed method.
3. The novelty of the method is to leverage attention mechanism to obtain the approximated correlation between labeled and unlabeled nodes instead of heuristic methods. While I am not an expert of active learning domain, I assume the direction of replacing heuristic to learned measure should be a proper path to better quantify the most quality label sets.

**Weaknesses:**

1. While the paper provides a comprehensive experiments over many datasets, as I am not an expert of active learning field, I found it rather confusing to use hyperparameter to search for an initial set S0. From my perspective, the initial labeled set S0 should be within part of the budget B. If hyperparameter search is allowed for S0, I think that implicitly provides exceeding budget to the proposed methods. I assume only the proposed method has such hyperparameter and therefore this leads to a unfair comparison and makes me doubt if the experimental advantage comes from the exceeding budget.
2. I am not really convinced on the theoretical analysis. In the original CNN paper regarding coreset analysis, the paper explicitly mentioned the data is iid, which is not the case for the graph data as the paper discussed in the introduction. Graph data is non-iid and there is no attempts on resolving such issue in the theoretical analysis. Proof of the proposition 4.1 is also not satisfying. for d(h,hv)<d(h',hv)<rv, what exactly do we know about h and h' in this case. Simply saying that h' = h+d doesn't provide any useful information for obtaining Eq 11. One can simply consider otherwise h = h'+d and obtain a reversed relation for Eq 11. Also, the paper states according to assumption 1, the dL/dh part is basically 0, then Eq 11 really doesn't provide any information from what I understand.
3. There exist multiple typos such as line 781 assumpation ; proposation, line 772 double equation, etc. The notation of the symbol is also very confusing, d is used both as a distance function, derivative, and variable in the proof, even in the same line. all of these lead to bad experience of reading.
4. There lacks a justification over why GCN, GAT and APPNP is used for graph aggregation but not other models, normally common choice should include GraphSage, GIN, etc. I don't find a clear explanation over why these models are selected as backbones.

**Questions:**

See Weaknesses, especially point 1 and 2.

---

> ### Author Response · Authors · 2024-11-23
>
> Dear Reviewer,
>
> We sincerely appreciate the your time and effort in providing valuable feedback. Below, we have addressed each of the concerns and questions raised.
>
> **Q1.**
> While the paper provides a comprehensive experiments over many datasets, as I am not an expert of active learning field, I found it rather confusing to use hyperparameter to search for an initial set S0. From my perspective, the initial labeled set S0 should be within part of the budget B. If hyperparameter search is allowed for S0, I think that implicitly provides exceeding budget to the proposed methods. I assume only the proposed method has such hyperparameter and therefore this leads to a unfair comparison and makes me doubt if the experimental advantage comes from the exceeding budget.
>
> **A1.** Please allow us to clarify the setting of initial set $S^0$:
>
> **1. Initial Set Within Budget:** It is correct that the initial labeled set $S^0$ should be part of the overall budget $B$. In our experiments, although we perform a search to determine $S^0$, we ensure that its size remains smaller than the final budget $B$. This ensures consistency across all baseline methods, where the same size of the training set is used to enable a fair comparison.
>
> **2. Purpose of Initial Set Search:** The search for the initial set $S^0$ is specifically designed to facilitate effective training for the subsequent coreset selection process, i.e., attention-based message-passing and data selection. The initial labels in $S^0$ are chosen to provide sufficient supervision for early training stages, ensuring that reliable attention coefficients are obtained. These coefficients are essential for capturing the global influence between labeled and unlabeled nodes, guiding the selection of informative data points from the unlabeled pool. In our experiments, we observed that setting $S^0 = 0.3*B$ is effective for commonly used datasets, enabling the selection of a representative coreset from the unlabeled data.
>
> **Q2.**
>  I am not really convinced on the theoretical analysis. In the original CNN paper regarding coreset analysis, the paper explicitly mentioned the data is iid, which is not the case for the graph data as the paper discussed in the introduction. Graph data is non-iid and there is no attempts on resolving such issue in the theoretical analysis. Proof of the proposition 4.1 is also not satisfying. for d(h,hv)<d(h',hv)<rv, what exactly do we know about h and h' in this case. Simply saying that h' = h+d doesn't provide any useful information for obtaining Eq 11. One can simply consider otherwise h = h'+d and obtain a reversed relation for Eq 11. Also, the paper states according to assumption 1, the dL/dh part is basically 0, then Eq 11 really doesn't provide any information from what I understand.
>
> **A2.** We appreciate the opportunity to clarify the points raised regarding theoretical analysis, especially Proposition 1 and its implications. Below, we provide further explanations to resolve any misunderstandings:
>
> **1. Applicability of Proposition 1 to Non-i.i.d. Graph Data:**
>
> Proposition 1 is formulated for abstract points in the embedding space, where the distances $d\left(u, v\right)$ and $d\left(u^{'}, v\right)$ are calculated based on the learned embeddings.
>
> Importantly, because Proposition 1 operates in the embedding space, it is agnostic to whether the original data is i.i.d. or non-i.i.d. In the case of graph data, the embeddings encode dependencies arising from graph structures. Thus, while the input graph data is non-i.i.d., the results of Proposition 1 remain valid, as they are based on the transformed embedding space, which already incorporates graph-specific information.
>
> We will explicitly mention in the revised manuscript that Proposition 1 is suitable for both i.i.d. data and non-i.i.d. graph data, as it relies on the embedding space rather than the raw input data.
>
> **2. Clarification of the Constraints $h = h' + d$:**
> Based on the condition $d(h, h_v) \leq d(h', h_v) \leq r_v$, we deduce that $h$ is closer to $h_v$ in the embedding space, with $h_v$ as the center. This implies that $h' = h + d$, where $d > 0$ represents a positive perturbation along the distance metric.
>
> The reversed case, $h = h' + d$, is not valid in this context because it would contradict the ordering of distances, i.e., $d(h, h_v) < d(h', h_v)$. We will explicitly add the constraint $d > 0$ to avoid ambiguity and clearly explain why the relationship $h' = h + d$ holds under this condition.

---

> > ### Author Response · Authors · 2024-11-23
> >
> > Thanks for your time! Here is the rest part.
> >
> > **Q3.**
> > There exist multiple typos such as line 781 assumpation ; proposation, line 772 double equation, etc. The notation of the symbol is also very confusing, d is used both as a distance function, derivative, and variable in the proof, even in the same line. all of these lead to bad experience of reading.
> >
> > **A3.**
> > Thank you for pointing out the typographical issues and inconsistencies in the notation. We apologize for the confusion caused by these errors. We will carefully check the whole paper in the final version.
> >
> > **Q4.**
> > There lacks a justification over why GCN, GAT and APPNP is used for graph aggregation but not other models, normally common choice should include GraphSage, GIN, etc. I don't find a clear explanation over why these models are selected as backbones.
> >
> > **A4.**
> >
> > **General Backbone Selection:** We would like to emphasize that the backbone used in AGCL can be any general GNN model. In our experiments, we chose GCN, GAT, and APPNP as example backbones because they are widely used in the literature. The proposed method, AGCL, is designed to be independent of the specific GNN architecture. Its primary focus is on selecting core data points from the unlabeled pool for training. Once these core data points are selected and labeled, they can be used for subsequent training with any backbone model of choice. This flexibility ensures that AGCL is applicable to a wide variety of GNN models, including but not limited to GCN, GAT, and APPNP.
> >
> > **Evaluation with GraphSAGE:** In response to your concern, we have conducted additional experiments to evaluate AGCL with GraphSAGE. The results, shown in the following table, indicate that AGCL performs consistently well across different GNN backbones. Specifically, AGCL outperforms other methods when using GraphSAGE, further demonstrating its versatility and effectiveness in core data selection, irrespective of the underlying GNN model.
> >
> > **Table: Classification accuracy (\%) of GraphSage on three citation datasets with different training sets.**
> > | **Methods**      | **Training data** | **Cora**         | **Citeseer**     | **Pubmed**       |
> > |------------------|-------------------|------------------|------------------|------------------|
> > | **GraphSage**    | GRAIN             | 81.55 ± 0.50     | 71.06 ± 0.44     | **79.23** ± **0.30** |
> > |                  | GraphPart         | 81.27 ± 0.43     | 70.42 ± 0.58     | 77.29 ± 0.35     |
> > |                  | **AGCL**          | **82.56** ± **0.39** | **72.64** ± **1.11** | 79.14 ± 0.33    |

---

> > ### Comment · Reviewer_Shvb · 2024-11-24
> > **about the theoretical part**
> >
> > 1.Thanks for your clarification on the theoretical part. I think I was confused by the ∥d∥ = 0 in the beginning, I assume this is a typo that ||d|| should be some small value close to 0 and therefore you want to apply taylor expansion with small d?
> >
> > 2.I also assume in your appendix proof, for f(h), f() is any GNN network. You are assuming that both f() and L() is smooth to allow Eq(11) to be valid. Please correct me if I am wrong and you are referring otherwise, it's hard to follow and judge the correctness of a proof if you didn't define all the symbols clearly.
> >
> > 3. My confusion is: why d is small enough compared to h, as you suggest, d is the difference between h and h', I am not sure why you can assume that d is small enough to truncate all the higher order terms?

---

> > > ### Author Response · Authors · 2024-11-24
> > >
> > > Dear Reviewer,
> > >
> > > Thank you very much for your quick response. We truly appreciate your careful reading and insightful questions.
> > >
> > > **Q. $||d|| = 0$**
> > >
> > > **A.**
> > > $||d|| = 0$ is actually a typo. The correct constraint with $||d|| > 0$ with $d<r_v$, where $r_v$ is the width of the neighbourhood in the embedding space that displays simple monotonic behaviour. $r_v$ can be viewed as the radius around the center $h_v$ (a visualization of the radius in the embedding space is provided in Fig. 1(a)).
> > >
> > > **Q. About $f(h)$ and $\mathcal{L}()$.**
> > >
> > > **A.**
> > > Your are correct. Here, $f(h_v)$ represents the prediction function applied to the embedding $h_v$ where $h_v = \mathcal{M}(v)$, and $\mathcal{M}$ denotes the GNN model.
> > > $\mathcal{L}(\cdot)$ is the loss function. Both $f(\cdot)$ and $\mathcal{L}(\cdot)$ are smooth.
> > >
> > > **Q. why d is small enough compared to h, as you suggest, d is the difference between h and h', I am not sure why you can assume that d is small enough to truncate all the higher order terms?**
> > >
> > > **A.**
> > > Thanks for your question.
> > >
> > > We assume that both $h$ and $h^{\prime}$ lie within the embedding space centered at $h_v$, satisfying $d\left(h, h_v\right) \leq d\left(h^{\prime}, h_v\right) \leq r_v$, where $r_v$ defines the width of the neighborhood around $h_v$. If $h$ or $h^{\prime}$ extend beyond this neighborhood (i.e., $h-h_v$ or $h^{\prime}-h_v$ becomes large or the difference between $h$ and $h^{\prime}$ is large), then they are not suitable for this.
> > >
> > > Under this assumption, we focus on the low-order terms for simplicity.
> > >
> > > Proposition 4.1 indicates that the closer the unlabeled data is to the labeled data in the embedding space, the smaller the loss for the unlabeled data. We will update the proof of Proposition 4.1 in the revised manuscript to provide clarification.
> > >
> > > We hope the updated details address your concerns effectively. Please feel free to reach out if you have any further questions or need additional clarification. We look forward to your feedback.

---

> ### Author Response · Authors · 2024-11-25
>
> Dear Reviewer,
>
> Thank you once again for your thoughtful and constructive feedback.
>
> In the updated manuscript, we have revised the proof of Proposition 4.1 and included additional experimental results on other GNN models (highlighted in blue). We hope these updates address your concerns and provide further clarity.
>
> As the discussion deadline approaches, we would greatly appreciate any additional suggestions you might have for improving our work. Your feedback is invaluable, and we sincerely welcome your guidance.
>
> If our responses have sufficiently addressed your concerns, we kindly hope you might reconsider the rating. Thank you again for your insightful review and thoughtful consideration.
>
> Best regards,
>
> The Authors

---

> > ### Comment · Reviewer_Shvb · 2024-11-25
> >
> > Dear author,
> > Thank you for your response. I think the current theoretical analysis is still overly simplified. Since the other part is fine to me, I will keep my current score.

---

> > > ### Author Response · Authors · 2024-11-26
> > >
> > > Dear Reviewer,
> > >
> > > Thank you for your follow-up and for taking the time to review our responses. We sincerely appreciate your acknowledgment of the other aspects of our work.
> > >
> > > **1.** Regarding the theoretical analysis, we would like to further illustrate why Proposition 4.1 is applicable to both CNNs and GNNs by providing a concrete example and additional explanations.
> > >
> > > By introducing the covering radius $\delta$ in [1], we theoretically show that for two testing samples, the one with a smaller distance to the training sample has better prediction performance in Proposition 4.1. Although the coreset analysis of [1] is specifically designed for CNN, the covering radius can be used in the embedding space for GNNs. For example, consider SGC [2], a GNN with a decoupled propagation mechanism. After feature propagation, only one MLP layer is used for prediction. Since the features are updated prior to the network, SGC can be viewed as a simplified CNN. Consequently, Proposition 4.1 is applicable to both i.i.d. data (CNNs) and non-i.i.d. graph data (GNNs).
> > >
> > > Furthermore, Theorem 1 in FeatProp [3] also indicates that the classification loss of GNN is informally bounded by the covering radius of the labeled set.
> > >
> > > **2.** While Proposition 4.1 establishes the foundation for Lemma 4.1, we emphasize that the most critical theoretical support for AGCL lies in Lemma 4.2. This lemma demonstrates the connection between coreset analysis (directly linked to prediction loss) and the attention-based scoring mechanism.
> > >
> > > Specifically, we show the advantage of selecting the coreset based on an explicit metric (representation scores) that quantifies the relationship between labeled and unlabeled data pools, leading to improved prediction accuracy.
> > >
> > > Building on Lemma 4.1 and Lemma 4.2, Theorem 4.1 states that in data selection, we need to explore samples in the unlabeled pool that have the maximum representation difference from the existing labeled pool in the well-defined representation space.
> > >
> > > According to Theorem 4.1, **1)** we need to explicitly show the relationship/distance/influence of unlabeled data and labeled data through a measure that relates to final prediction performance, and **2)** the measure function needs to capture both feature and complex graph structural information from a global perspective.
> > >
> > > We hope these additional clarifications address your concerns about the theoretical analysis. If you have any further suggestions or specific recommendations, we would greatly appreciate your input.
> > >
> > > Thank you for your time and thoughtful consideration.
> > >
> > > [1] Sener, Ozan, and Silvio Savarese. "Active learning for convolutional neural networks: A core-set approach." ICLR 2018.
> > >
> > > [2] Felix Wu, et al. "Simplifying graph convolutional networks." ICML, 2019.
> > >
> > > [3] Yuexin Wu, et al. "Active learning for graph neural networks via node feature propagation." arXiv preprint arXiv:1910.07567 (2019).
> > >
> > >
> > > Best regards,
> > >
> > > Authors

---

### Meta-Review · Area_Chair_4UU6 · 2024-12-16

**Metareview:**

This paper proposes an attention-based graph coreset labeling framework to explicitly explore the influence of labeled data on unlabeled data. Rather than implicit or heuristic based approach such as entropy or graph-based information to select labeled node from unlabeled set, it explicitly explores and exploits the correlations between nodes in the unlabeled pool and those in the labeled pool using an attention architecture and directly connects the correlations with the prediction performance on unlabeled set. Experimental results demonstrate its effectiveness.


After rebuttal, two out of four reviewers are still negative about this manuscript, and hold that the theoretical analysis is overly simplified, and the comparison methods used in the paper are somewhat outdated. Besides, there exist some ambiguous definitions, and the first assumption about the first order taylor approximation is not fully convincing. So, more efforts could be needed.

**Additional Comments On Reviewer Discussion:**

Two out of four reviewers are still negative about this manuscript, and hold that the theoretical analysis is overly simplified, and the comparison methods used in the paper are somewhat outdated. Besides, there exist some ambiguous definitions, and the first assumption about the first order taylor approximation is not fully convincing.

---

### Decision · Program_Chairs · 2025-01-22

Reject